# Misinformation about medication during the COVID– 19 pandemic: A perspective of medical staff

**Claudiu Coman**[1]*, **Maria Cristina Bularca**[1], **Angela Repanovici**[2], **Liliana Rogozea**[3]

1 Department of Social Sciences and Communication, Faculty of Sociology and Communication, Transilvania University of Brasov, Brasov, Romania, 2 Department of Product Design, Mechatronics and Environment, Faculty of Product Design and Environment, Transilvania University of Brasov, Brasov, Romania, 3 Basic, Preventive and Clinical Sciences Department, Transilvania University of Brasov, Brasov, Romania

* claudiu.coman@unitbv.ro

## Abstract

### Background

Healthcare professionals had to face numerous challenges during the pandemic, their professional activity being influenced not only by the virus, but also by the spread of medical misinformation. In this regard, we aimed to analyze, from the perspective of medical staff, the way medical and non—medical information about the virus was communicated during the pandemic to encourage the development of future research or interventions in order to raise awareness about the way misinformation affected medical staff.

### Methods and findings

The study was conducted on Romanian healthcare professionals. They were asked to answer to a questionnaire and the sample of the research includes 536 respondents. The findings revealed that most respondents stated that information about alternative treatments against the virus affected the credibility of health professionals, and that younger professionals believed to a greater extent that trust in doctors was affected. The research also showed that respondents were well informed about the drugs used in clinical trials in order to treat the virus.

### Conclusions

Healthcare professionals declared that the spread of misinformation regarding alternative treatments, affected their credibility and the relationship with their patients. Healthcare professionals had knowledge about the drugs used in clinical trials, and they acknowledged the role of social media in spreading medical misinformation. However, younger professionals also believed that social media could be used to share official information about the virus.

**Data Availability Statement:** All relevant data are within the paper and its Supporting Information files.

**Funding:** The authors received no specific funding for this work.

**Competing interests:** The authors have declared that no competing interests exist.

## Introduction

The COVID 19 pandemic generated multiple changes in the way today's society members carry out their daily activities. One of the processes which was mostly affected by the pandemic was the communication process between institutions and the public, as well as between individuals. In this regard, from this perspective, while many domains were affected by the spread of the virus, such as the educational system or the cultural sector, the health sector was the one that faced the most challenges, [1].

"Caused by severe acute respiratory syndrome coronavirus 2 [2], the disease was firstly detected in December 2019, in Wuhan, China [3]. Due to the evolution of the virus, the World Health Organization declared the pandemic in March 2020 [4], and as of November 27 over 61 million cases were reported [5]. In this regard, although several companies are struggling to develop a vaccine, and some of the proposed vaccines showed promising results [6], so far no vaccine was approved in order to be administrated to the entire population [7]. Ever since the pandemic was declared, many companies started to be preoccupied with finding a treatment, and one method used that was adopted was administrating to patients, drugs that were previously used for curing other viruses [8]. Thus, one of the most well—known trials started was the SOLIDARITY trial, which focused on using various drugs including chloroquine and hydroxychloroquine, lopinavir or ritonavir [9]. However, even if those drugs were taught to have positive effects on treating the virus, they did not have a significant influence on preventing mortality in general [10].

With the development of many trials and programs meant to find a cure for COVID 19 and with the use of diverse drug combinations, another major problem arose: misinformation and fake news about the virus, its treatment or methods to combat it. In this regard, along with the pandemic, people also had to face an epidemic of information, described by the general director of WHO as an „infodemic" [11]. In other words, information about COVID 19 began to be spread by people on every available communication channel, both in the online and offline environment. However, very often and especially on social media, the information was poorly communicated, it was distorted and there usually wasn't enough scientific evidence to demonstrate its validity [12].

Taking into account the previously mentioned aspects the paper addresses the issues of drugs tested and used for the treatment of COVID 19 and how information about COVID 19 was communicated in the offline and online environment. The purpose of the paper is to analyze, from the perspective of medical staff, the way medical and non—medical information about the virus was communicated during the pandemic in order to encourage the development of future research or interversions in order to raise awareness about the way misinformation affected medical staff. Thus, the paper aims at finding an answer to three research questions: (1) to what extent information about alternative treatments affected the credibility of medical staff? (2) What is the knowledge of medical staff about the type of drugs that had positive effects on treating the disease and about alternative treatments? (3) How satisfied is the medical staff with the way medical and non-medical information was communicated online and offline during the pandemic? (4) What is the perception of medical staff about the role of social media in spreading misinformation about the virus? (5) What aspects of the professional activity of the medical staff were affected most by the COVID– 19 pandemic?

## Literature review

### Information on drugs used to treat COVID 19

Before analyzing the way information about the virus was communicated in the online environment, it is important to take a look at the drugs used to treat the disease. Hence, one of the

most important issues that appeared with the COVID 19 pandemic, was finding the right treatment for the virus. In this regard, researchers started to develop many experimental trials and used diversified drug combinations in order to treat patients with COVID 19. However, information that was communicated about the effectiveness of certain drugs was often contradictory.

Chloroquine and hydroxychloroquine are two drugs that were tested and included in many trials. Both drugs were previously used to treat malaria but they also have antiviral effects on viruses like HIV since they have the ability to prevent the virus to enter in the host cells [13]. Even though they have similar compounds, chloroquine is taught to have more negative effects than hydroxychloroquine [14], and hydroxychloroquine is considered safer due to the fact that it can be tolerated better for a longer period of time [15].

While some studies show positive effects of hydroxychloroquine in inhibiting the infection with the virus in vitro [16, 17], other studies found no influence of the drug on mortality rate or time spent by patients in the hospital [18]. However, when hydroxychloroquine was combined with other drugs such as azithromycin, it showed beneficial effects in treating patients with COVID 19 [19].

Nonetheless the findings regarded the effectiveness of these drugs were contrasting. For example, on March 28 2020 the Food and Drug Administration (FDA) issued an Emergency Use Authorization for using hydroxychloroquine in treating people suffering from COVID 19 [20], and in June 15 2020, the FDA retracted the authorization stating that the trials in which the drug was involved showed that the drug had no effect on the faster recovery of patients or on decreasing chances of death [21]. Even more, on 5th June 2020 the UK trial, Randomised Evaluation of COVID 19 THERAPY (RECOVERY), also stopped testing the drug on patients because the results showed no benefits in improving the conditions of hospitalized patients with COVID 19 [22].

Studies were carried out with other drugs such as lopinavir/ritonavir, an antiviral drug used in the treatment of HIV [23]. While in concentration of 4 μg/ml and 50 μg/ml, the drug showed positive effects against the virus in vitro [24], a study on 199 patients, from which 99 received the drug and the other 100 did not receive the drug, revealed that lopinavir/ritonavir had no benefits when it comes to diminishing mortality or improving the state of patients with severe symptoms [25].

Controversial discussions also involved the use of Ibuprofen, a Non-steroidal anti-inflammatory drug that is used to treat fever, or inflammation [26]. Since the pandemic was declared there has been a preoccupation regarding ibuprofen and its role in making people more vulnerable to contacting the virus. Thus, right after the declaration of the pandemic, in a letter addressed to The Lancer Journal, researchers pointed out that ibuprofen could make people with diabetes, cardiac disease or hypertension more likely to get infected with virus and have severe symptoms [27]. However, while firstly, WHO recommended people who are infected with the virus not to take ibuprofen, only one day after that recommendation, on 18 March 2020, WHO corrected its statement and mentioned that it"does not recommend against ibuprofen" [28]. Even more, a study focusing on the use of ibuprofen showed that the drug does not make patients feel worse [29] and another study that analyzed the use of ibuprofen and paracetamol of 403 COVID 19 confirmed patients revealed that compared to paracetamol, ibuprofen did not aggravated the clinical state of the patients [30].

While other drugs failed to show beneficial effects on the treatment of COVID 19, drugs like dexamethasone, which is included in the UK RECOVERY trial, revealed positive effects on people suffering from COVID 19: the drug lowered the risk of death in patients on ventilators from 40% to 28% and in patients who were in need of oxygen, from 25% to 20%, but did not influence the state of patients who did not need oxygen [31, 32].

Another highly tested drug was Remdesivir, an antiviral drug produced by Gilead Sciences that was previously used in treating Ebola [33]. The information regarding its positive effects on treating COVID 19 is also contradictory. A study conducted from February 6 2020 until March 12 2020, on 237 patients, showed that the drug did not bring any benefits for people that had severe symptoms of COVID 19 [34], while a more recent study revealed that Remdesivir had a more positive effect in reducing the time of recovery in patients with COVID 19 that showed signs of respiratory issues, than it had the placebo effect [35]. However, the FDA approved on October 22 2020, the use of Remdesivir in the case of adults and also children aged 12 or older who have at least 44 kilograms, who are infected with the virus and need to be treated in the hospital [36], and as of November 20 2020, FDA allows, in emergency cases, the use of Remdesivir in combination with Baricitinib, for adults and children aged two or older that require oxygen and treatment in the hospital [37].

## Social media and COVID 19 misinformation

Together with the health crisis, the COVID 19 pandemic generated an information crisis, often described as an infodemic, that is represented by the spread of fake news, misguided and false information, especially in the online environment [38].

In this context, social media plays an essential role in disseminating information. Social media consists of internet based channels that provide people with the opportunity to interact, communicate in asynchronous way and in real time, with either small or large audiences where value is derived from user generated content [39]. Social media comprises multiple social networks, which according to Boyd and Ellison, offer users the possibility to create profiles that are public, or semi-public, to create a list of people with whom they can interact and share information and to view the list of connections that other users make [40].

Social media channels are often used in time of crisis not only by citizen, but also by official authorities, emergency services, because they can facilitate communication and the spread of valuable information that can contribute to surpassing the crisis [41]. Social networks like Facebook, Whatsapp, Twitter, Instagram can function as sources that have the ability to confirm or complete the information communicated by the authorities, while also receiving feedback from the public [42]. Thus, sending messages through social media channels is a strategy that can help authorities obtain feedback on certain proposals regarding public health policies [43]. Even more, a study regarding the influence of social media on the way people protect their health during the pandemic, showed that social media can have positive impact on increasing awareness about public health and protection against the virus [44].

However, during the pandemic, while authorities can use social media to keep the public informed, a major issue generated by social media, that public health representatives have to face, is the spread of fake news [45].

Fake news are represented by fabricated information designed in the form of news communicated by the media that do not share the same process of organization and do not have the same intent, and fake news are related to misinformation: information that is false or misleading, and disinformation: a type of false information whose aim is to deceive people [46].

Thus, the internet became a favorable environment for spreading conspiracy theories or false information about alternative treatment for the virus. Since people were stressed and frightened by the uncertainty of the situation, they started to consider reasonable and valid any information that presented explanations in regards to the virus [47]. Thus, when referring to health information, false news often undermine the credibility of official sources, they create confusion among people and favor the faster spread of the virus [48].

Misinformation during the pandemic can negatively influence peoples' health because false information is not easy to recognize, because it can determine people to change their behavior

in a way that is harmful to their health and those around them. Thus, since the pandemic was declared, false information has been spread about the origin of the virus, about what caused it, how it spreads and what treatment is efficient for eliminating it [49]. However, a study focusing on the WhatsApp platform showed that when the information on social media is shared by trusted sources, it can increase knowledge about the virus and encourage people to adopt preventive behavior [50].

During the time of crisis, on platforms like WhatsApp or Facebook, more and more false news and unverified information about the virus began to be shared. With millions of users worldwide, WhatsApp became one of the platforms where most fake news were shared by forwarding messages to many users [51], while Facebook was characterized as the core, epicenter of misinformation [52].

When it comes to health misinformation on social media, the most discussed subjects are alternative cures involving certain food or drinks, hygiene related actions and treatment drugs. Thus, among the most "recommended" practices for preventing or curing COVID were drinking hot water every 15 minutes in order for the virus to go into the stomach, eating garlic, taking vitamin C or even pointing a hairdryer to the nostrils because the heat could eliminate the virus [53].

False news that circulated on social media regarding the virus also involve the idea that the virus was created on purpose in a lab, three in ten Americans considering true this information [54].

However, many other unverified methods were shared and the most forwarded messages on WhatsApp presented information about the fact that if people hold their breath for ten seconds without coughing then they are not infected with the virus, about the idea that at temperatures of 30–35 Celsius degrees the virus will die, messages about the release of the vaccine or about drugs allegedly recommended by Chinese doctors that could be efficient in eliminating the virus [55].

Nonetheless, misinformation became a major issue in the context of the pandemic, but also a subject of interest for researchers. A study focusing on the spread of fake news showed that most news reconfigure and twist the original information thus creating a different context, and that most of them contain false information about public authorities and health organizations [56].

Another study found that people who tend to rely on their intuition or who possess little scientific knowledge about certain subjects, encountered difficulties in differentiating true and false information [57]. Thus, misleading or unverified information can negatively influence the way people behave. For example, people in USA who died after they consumed chloroquine may have used the drug because news about it mentioned that it could treat and eliminate the virus [58]. Even more, a study concerning misinformation on Facebook revealed that posts made from verified accounts contained more false information than the accounts that were not verified [59], while other study conducted from 23 April 2020 to 27 April 2020, focused on perception about contradictory information and stated that 73% of participants mentioned they observed or were exposed to contrasting messages usually communicated by politicians or health experts [60].

Apart from influencing peoples' beliefs or health practices, COVID 19 fake news also influenced the activity of health professionals. Social media managed to increase the level of trust in information that comes from people's personal opinions rather than professionals [61], and doctor's credibility is often affected. In order to improve these situations, doctors must be willing to use social media not just to send messages, but to actively communicate with people, to offer feedback, to share their experiences and rectify and clarify the fake news presented on social media [62].

Among action from health professionals, in order to combat COVID 19 fake news, social media networks as well as public authorities must implement some strategies. For example, the government of United Kingdom developed collaboration programs between its rapid response teams and social media platforms, and Taiwan introduced greater fines for news that were proven to be false [63]. Moreover, even though some social networks such as Facebook or Twitter already implemented algorithms to identity and remove fake accounts [64], or to correct information [65], they should further develop efficient strategies in order to validate the information that people share [66].

## The influence of the pandemic on doctors' credibility and relationship with patients

The way information regarding the virus was communicated online and offline during the pandemic played an essential role in the process of maintaining trust in health professionals. In this regard, a previous longitudinal study conducted in Poland revealed that trust in physicians has declined from 2018–2020, and emphasized the idea that the decrease may be caused by the health problems that people had to cope with during the pandemic and the problems with the healthcare system of the country [67]. In Romanian context, a previous study showed that the communication process of the healthcare system was poor and confusing, and that public health authorities at national level focused more on global information about the virus, while local authorities failed to succeed in providing their "share of information" [68]. Another study, which focused on analyzing the online communication of Public Health Agencies from Italy, United States and Sweden, revealed that compared to Sweden and the United States, agencies from Italy collaborated more with other organizations, and that overall, the communication process of the agencies was coordinated by their members, that agencies also communicated with governments, but they rarely collaborated with political or non-governmental organizations [69]. Hence, while trust in the government and communication from authorized organizations is essential, the importance of trusting the professionals is highlighted by a study conducted in Thailand, which showed that in the cases in which people have low levels of trust in the government, trust in professionals can have a positive influence on the adoption of protective measures at the individual level [70].

Furthermore, another previous study conducted in Poland, revealed that information can have the power to influence the level of trust that people have in the healthcare system and in healthcare professionals, suggesting that an increase of trust in hospitals, may be associated with a decrease of trust in physicians [71].

While focusing on studying people's response to non- pharmaceutical interventions, conspiracy theories and alternative treatments, a study conducted in Finland showed that the level of trust people have in the system implemented in order to provide information about the virus, has an essential role in the way people react to the official measures recommended. Hence, most participants in the study were between 40 and 60 years of age, and the study emphasized that people who were less willing to comply with the non-pharmaceutical interventions implemented by the government, tended to believe more in conspiracies and had low levels of trust in the sources which provided information about the virus [72].

Another study, which focused on examining the relationship between trust in the healthcare system and people's choice of seeking medical help when they experienced COVID– 19 symptoms, concluded that high levels of trust in the healthcare system can increase the probability of asking for medical help when people first notice COVID– 19 symptoms [73].

Taking into account the aspects mentioned above, we can infer that peoples' trust in doctors was affected during the pandemic. In this regard, in the context of misinformation, one of the

reasons why people lost trust in doctors may be the fact that, besides using social media for communicating information, for networking or for interacting with patients, many medical or dental practitioners used social media to express their professional opinions about the virus, opinions which were not validated and which later proven to be inaccurate [74]. In other words, health professionals may have contributed to the spread of misinformation, and such behavior can contribute to the decrease of trust in medical processes and in healthcare professionals [75]. Other researchers who focused on examining medical misinformation, found that most doctors (94.2%) stated that patients had medical misinformation, and the subjects about they had the most inaccurate information were represented by COVID– 19 vaccines, COVID– 19 origin, treatment or essential oils [76]. Furthermore, a previous study discovered that trust in doctors increased with age, and communication difficulties decreased, and that trust in doctors decreased while the level of education and communication difficulties increased [77].

Hence, while acknowledging that the pandemic influenced the trust in medical professionals, another aspect that was negatively influenced was the relationships between doctors and their patients. A study which focused on examining the doctor–patient interaction from the perspective of both groups of people, revealed differences in the respondents' opinions. Thus, most doctors stated that they still make eye contact (72%) and that they still show patients empathy, but only few patients declared that their doctors made eye contact (56,8%) or showed them empathy (43,2%) [78].

## Materials and methods

### Research design

The present study was conducted on Romanian healthcare professionals including doctors, nurses and medical students. The method used is quantitative and descriptive The questionnaire was administrated online, the data was collected through the help of Google forms, and was disseminated on groups of healthcare professionals and students on platforms such as Facebook and WhatsApp, during the period April 2021– June 2021. The data we collected was firstly exported to Microsoft Excel, and then it was analyzed with IBM Statistical Package for the Social Sciences, version 20. The respondents were informed about the purpose of the study, about the fact that they were allowed to withdraw at any time, and they were asked to give their consent for participating in the study. The average time needed to complete the questionnaire was 15 minutes.

Considering the validity of our research, we took into account the theoretical information from the literature regarding the development of a questionnaire. Our team of researchers together with health specialists have configured the dimensions, and operationalized the concepts in accordance with the theoretical approaches identified at the current stage of the research. Even more, we pre-tested the questionnaire before disseminating in order to guarantee the validity of the instrument. Thus, the questionnaire was completed by 50 respondents in the pre-testing stage. Considering the reliability of the research, we used split half reliability method. We split our sample in half, and we checked the variables in from our sub-samples in order to see if the variables provided convergent results. The convergent results we obtained by applying the split half method showed that we obtained a high fidelity measurement.

### The research instrument

In order to conduct the research we used a quantitative method while having a questionnaire as an instrument. In this regard, we developed a questionnaire which comprises four sections: A. Influence of the pandemic on the professional activity of medical staff (items A1 to A4), B. Perception about the authorities' communication process (items B1 to B11), C. Perception

about the communication of non- validated treatments (items C1 to C20), and D. Sociodemographic questions (items D1 –D9), such as: gender, age, living environment, professional degree, field of specialization. The sociodemographic questions were used in order to identify different or similar attitudes between specific groups. The questionnaire can be found in "S1 Appendix English version of the questionnaire", and in "S2 Appendix Romanian version of the questionnaire." Before disseminating the questionnaire, the instrument was tested on 30 doctors who work in the field of cardiology and general medicine. The respondents understood clearly the questions and did not report any issue in the process of answering them. Hence, the questionnaire comprises close ended and open ended questions (Items A1, A4,B3, B11, C19, C20, D2, D5, D6,) dihotomic questions as well as questions whose answers were measured on a 7 point Likert scale. For example, item A2 measured the extent to which the respondents considered that the pandemic influenced the way they carried out their professional activity (1- "to an extremely little extent, 7 "to an extremely great extent"), or item B2 measure the respondents' level of agreement with statements regarding the way authorities communicated during the pandemic (1 –"strongly disagree, 7-"strongly agree").

## Sampling and data collection procedures

In order to conduct the research we used a quantitative method while having as an instrument a questionnaire. The responses were collected online, with the help of Google forms, and the questionnaire was self–administrated. The research received approval from The Council of the Faculty of Sociology and Communication, approval request Nr.378/30.03.2021. Taking into account the sampling method and the calculation of the study sample, we used random, probabilistic sampling method. We took into consideration specialists, physicians,and medical students from Brasov, and we applied the snowballing method in order to disseminate the questionnaire. The sample of our study comprises 536 respondents, and includeds doctors, nurses as well as medical students from Romania.

## Data analysis

Data was analyzed with IBM Statistical Package for the Social Sciences, version 20. In order to analyze the data and identify differences and similarities between the attitudes of certain groups, t tests for independent samples were performed. The t test were performed among groups: male/female, working in unit with COVID– 19 patients/ not working in unit with COVID– 19 patients, urban/rural area, and professional degree: medical staff/students. Hence, in order to be able to analyze the results depending on professional degree, we computed the variable of professional degree which had the following values: senior specialist medical–doctor, specialist medical–doctor, resident, nurse with higher education diploma, nurse with other studies than higher education, medical student, student at university nursing program, in a new variable. Thus, doctors, nurses and residents, were integrated in a new group called "medical staff", while medical students and students at university nursing programs were integrated in the group "students". Moreover, for a better understanding of the way some variables correlate with each other, (for example: respondents satisfaction with the way authorities communicated during the pandemic and age, respondents' opinion about the way misinformation about alternative treatments influenced doctors' credibility and age), we also calculated the Pearson coefficient.

## Results

Out of the 536 respondents, 460 (85.8%) were female and 76 (14.2%) were male. A total of 411 respondents live in the urban area (76.7%), while 125 (23.3%) live in the rural area. Most respondents (286, 53.4%) are between 18 and 35 years of age, 142 respondents (26.5%) are

**Table 1. Sample characteristics (n = 536).**

|  | Category | Count | Percentage |
|---|---|---|---|
| Gender | Female | 460 | 88.8% |
|  | Male | 76 | 14.2% |
| Living environment | Urban | 411 | 76.7% |
|  | Rural | 125 | 23.3% |
| Age | 18–35 years old | 286 | 53.4% |
|  | 36–50 years old | 142 | 26.5% |
|  | 51–65 years old | 102 | 19.0% |
|  | Over 65 years old | 6 | 1.1% |
| Professional degree | Senior specialist medical—doctor | 102 | 19.0% |
|  | Specialist medical—doctor | 46 | 8.6% |
|  | Resident | 28 | 5.2% |
|  | Nurse with higher education diploma | 70 | 13.1% |
|  | Nurse with other studies than higher education | 48 | 9.0% |
|  | Medical student | 120 | 22.4% |
|  | Student at university nursing program | 122 | 22.8% |
| Field of specialization | General medicine | 378 | 70.5% |
|  | Family doctor | 56 | 10.4% |
|  | Pediatrics | 16 | 3% |
|  | Stomatology | 10 | 1.9% |
|  | Oncology | 10 | 1.9% |
|  | Surgery | 8 | 1.5% |
|  | Internal medicine | 8 | 1.5% |
|  | Virology/ infectious disease doctor | 6 | 1.1% |
|  | Cardiology | 6 | 1.1% |
|  | Radiology | 6 | 1.1% |
|  | Other | 32 | 6% |
| Works in a unit with COVID– 19 patients | Yes | 122 | 22.8% |
|  | No | 414 | 77.2% |

between 36 and 50 years of age, 102 respondents (19.0%) are between 51 and 65 years of age, and 6 of them (1.1) are over 65 years of age. When it comes to the professional degree of the respondents, most of them are students at a university nursing program (122, 22.8%), and medical students (120, 22.4%). However, a total of 102 respondents (19.0%) are senior specialists medical–doctors, and 70 (13.1%) are nurses who have a higher education diploma. When it comes to the respondents field of specialization, most of them (70.5%) operate in the field of general medicine, while others are family doctors (10.4%), pediatricians (3%), dentists or oncologists (1.9%), surgeons of doctors who are specialized in internal medicine (1.5%), or infectious disease doctors, radiologists or cardiologists (1.1%). Furthermore, most of the respondents (77.2%) stated that they did not work a unit with COVID– 19 patients while few of them (22.8%) stated that they worked in such a unit at the time the research was conducted. Thus, all the characteristics of the sample are presented in Table 1.

## 1) To what extent information about alternative treatments affected the credibility of medical staff?

The results of our research revealed that respondents were of the opinion that information about alternative treatments for COVID -19 affected the credibility of healthcare professionals. Hence, most respondents (32.5%), stated that trust in healthcare professionals was affected to a

**Table 2. The extent to which information about alternative treatments affected trust in physicians.**

| | | Frequency | Percent | Valid Percent | Cumulative Percent |
|---|---|---|---|---|---|
| Valid | to an extremely little extent | 14 | 2.6 | 2.6 | 2.6 |
| | to a very little extent | 10 | 1.9 | 1.9 | 4.5 |
| | to a little extent | 42 | 7.8 | 7.8 | 12.3 |
| | nor to a little, neither to a great extent | 58 | 10.8 | 10.8 | 23.1 |
| | to a great extent | 114 | 21.3 | 21.3 | 44.4 |
| | to a very great extent | 124 | 23.1 | 23.1 | 67.5 |
| | to an extremely great extent | 174 | 32.5 | 32.5 | 100.0 |
| | Total | 536 | 100.0 | 100.0 | |

an extremely great extent by the information about alternative treatments, many of them declared that credibility was affected to a very great extent (23.1%), and to a great extent (21.3%) (Table 2).

Furthermore, the Pearson correlation performed between the extent to which respondents believed that information about alternative treatments affected people's trust in doctors and the age of the respondents, revealed a weak, negative and statistically significant correlation between the two variables ($r(534) = -.155$, $p = 0.001$) (Table 3). Hence, as the age of the medical staff decreases, the extent to which they believe the credibility of doctors was affected increases. In other words, compared to older healthcare professionals, younger healthcare professionals tend to believe more that information about alternative treatments affected trust in doctors. One possible explanation for this result can be that younger people tend to be fonder of keeping up with trends and being up to date, and in this context, it is possible that they came into contact more frequently with information about certain alternative treatments for COVID– 19, this making them more aware about the way such treatments can undermine doctor's credibility.

In order to observe if there any differences in the opinion of the respondents depending on certain variables including, age, gender, or living environment, we performed t tests for independent samples. The results of the significant t tests (Table 4), showed that students believed to a greater extent (M = 5.60, SD = 1.49), that information about alternative treatments negatively affects the credibility of doctors, than the medical staff (M = 5.33, SD = 1.54). Also, respondents who declared they worked in a unit without COVID– 19 patients (M = 5.53, SD = 1.49), were more of the opinion that information about alternative cures affected trust in

**Table 3. Pearson correlation between information about alternative treatments and age.**

| | | C14. The extent to which information about alternative treatments affected trust in physicians | D2. Age |
|---|---|---|---|
| C14.[1] The extent to which information about alternative treatments affected trust in physicians | Pearson Correlation | 1 | -.155** |
| | Sig. (2-tailed) | | .000 |
| | N | 536 | 536 |
| D2[2]. Age | Pearson Correlation | -.155** | 1 |
| | Sig. (2-tailed) | .000 | |
| | N | 536 | 536 |

**. Correlation is significant at the 0.01 level (2-tailed).

[1] C14 –refers to the question 14 from the section C of the manuscript (The extent to which information about alternative treatments affected trust in physicians), section which refers to Perception about the communication of non- validated treatments

[2] D2—refers to question 2 from the D section of the manuscript (age), which refers to Sociodemographic characteristics of the respondents

Table 4. Significant t-test results: Comparisons between variables.

| | Group | N | Mean | S. D. | t-test for Equality of Means | | | | | | |
| | | | | | t | df | p | Mean Difference | Std. Error Difference | CI4 | |
| | | | | | | | | | | Lower | Upper |
| Variables: Information about alternative treatments _ Professional degree[1] | Medical staff | 294 | 5.33 | 1.54 | -2.04 | 534 | .04 | -.27 | .13 | -.52 | -.01 |
| | Student | 242 | 5.60 | 1.49 | | | | | | | |
| Variables: Information about alternative treatments _working unit | Unit with COVID -19 patients | 122 | 5.19 | 1.61 | -2.13 | 534 | .03 | -.33 | .15 | -.64 | -.02 |
| | Unit without COVID 19 patients | 414 | 5.53 | 1.49 | | | | | | | |
| Variables: Information about alternative treatments _gender | Male | 76 | 5.10 | 1.70 | -2.16 | 534 | .03 | -.40 | .18 | -.77 | -.03 |
| | Female | 460 | 5.51 | 1.48 | | | | | | | |

[1]Index variable from the professional degrees of respondents. Student: medical student and student at university nursing program, Medical Staff: Senior specialist medical–doctor, Specialist medical–doctor, Resident, Nurse with higher education diploma, Nurse with other studies than higher education

health professionals, than respondents who worked in a unit with COVID– 19 patients (M = 5.19, SD = 1.61). One possible explanation would be that, doctors who interacted with COVID– 19 patients may have observed that when being put in the situation to receive medical care in the hospital, patients still had faith and trust in doctors. Moreover, another explanation is that respondents who did not come into contact with COVID– 19 patients were not that close with the situation and thus they might have had a more distorted perception about the situation than those professionals who interacted with COVID– 19 patients. Moreover, the results of the research also showed that female respondents (M = 5.51, SD = 1.48), believed more than male respondents (M = 5.10, SD = 1.70), that trust in healthcare professionals was affected by the information about alternative treatments.

## 2) What is the knowledge of medical staff about the type of drugs that had positive effects on treating the disease and about alternative treatments?

Considering the type of drugs which were known to have positive effects on treating the virus, the research revealed that type of drug about which the respondents have heard it had positive effects against the virus was Dexamethasone (46.6%), closely followed by Remdesivir (40.5%) and Azithromicin (38.4%). However, some of the respondents also mentioned Chloroquine, Hydroxychloroquine (23.1%), Ibuprofen (19.8%), Tocilizumab (15.9%), and Favipiravir (13.8%) as drugs known to have positive effects when dealing with COVID– 19 (S1 Table with results to the 2nd research question_Table A). Hence, the research showed that the medical staff had knowledge about the type of drugs tested or used against the virus, which were taught to be efficient in treating the disease.

In the context of respondents' perception about alternative methods of preventing and treating the virus, the findings show that, most of them stated that they heard about the fact that alcohol consumption can prevent the infection with the virus (24.3%), that drinking warm water every 15 minutes may help eliminate the virus (21.3%), but also that pointing the hot air of the hairdryer to the nostrils leads to the elimination of the virus (16.8%) (S1 Table with results to the 2nd reseach question_Table B).

## 3) How satisfied is the medical staff with the way medical and non-medical information was communicated during the pandemic?

The findings of the study revealed that respondents were mostly dissatisfied with the way medical and non–medical information was communicated during the pandemic. Hence, the sum

**Table 5. The level of satisfaction with the way information about drugs used to treat the virus were communicated at national level.**

| | | Frequency | Percent | Valid Percent | Cumulative Percent |
|---|---|---|---|---|---|
| Valid | extremely dissatisfied | 52 | 9.7 | 9.7 | 9.7 |
| | very dissatisfied | 76 | 14.2 | 14.2 | 23.9 |
| | dissatisfied | 110 | 20.5 | 20.5 | 44.4 |
| | Nor dissatisfied, neither satisfied | 136 | 25.4 | 25.4 | 69.8 |
| | satisfied | 108 | 20.1 | 20.1 | 89.9 |
| | very satisfied | 30 | 5.6 | 5.6 | 95.5 |
| | Extremely satisfied | 24 | 4.5 | 4.5 | 100.0 |
| | Total | 536 | 100.0 | 100.0 | |

of the responses with negative valences of the study participants (extremely dissatisfied, very dissatisfied and dissatisfied), showed that 238 of them, (44.4%) were dissatisfied with the process of sending medical and non- medical information, while the sum of the positive responses (satisfied, very satisfied, extremely satisfied) showed that 162 of them (30.2%), were satisfied with the communication process (Table 5). In other words, the study highlighted that respondents registered mostly low level of satisfaction with the way information was sent during the pandemic.

Furthermore, in the context of the medical staff's satisfaction with the way information about drugs used to treat the virus was communicated at national level, the research showed that as age of the respondents decreases, the level of satisfaction increases ($r(534) = -.091$, $p = 0.035$) (Table 6). Thus, according to this result, it can be inferred that younger people were more satisfied than older people, with how information about drugs used to treat the virus was communicated.

Moreover, when asked to evaluate the efficiency of the communication strategies adopted by authorities in order to send information about the virus, most respondents stated that the strategies were effective. Thus, the sum of the responses with negative valences shows that 144 of them (26, 9%) described the communication strategies as inefficient, while 266 of them (49, 6%) described them as efficient (S2 Table with results to the 3rd research question_Table C). One interesting result of the analysis, was that, when trying to examine if the responses of the study participants about the efficiency of communication strategies differ depending on

**Table 6. Pearson Correlation: satisfaction with the way information about drugs used to treat the virus was communicated and age.**

| | | B10. Satisfaction with the way information about drugs used to treat the virus was communicated | D2. Age |
|---|---|---|---|
| B10[1]. Satisfaction with the way information about drugs used to treat the virus was communicated | Pearson Correlation | 1 | -.091* |
| | Sig. (2-tailed) | | .035 |
| | N | 536 | 536 |
| D2[2]. Age | Pearson Correlation | -.091* | 1 |
| | Sig. (2-tailed) | .035 | |
| | N | 536 | 536 |

*. Correlation is significant at the 0.05 level (2-tailed).

[1] B10- refers to the question 10 from the section B of the manuscript (Satisfaction with the way information about drugs used to treat the virus was communicated) section which refers to Perception about the authorities' communication process

[2] D2—refers to question 2 from the D section of the manuscript (age), which refers to Sociodemographic characteristics of the respondents.

**Table 7. Significant t test for information about drugs used to treat the virus and professional degree.**

| | | | | | t-test for Equality of Means | | | | | | |
|---|---|---|---|---|---|---|---|---|---|---|---|
| | Group | N | Mean | S. D. | t | df | p | Mean Difference | Std. Error Difference | CI4 | |
| | | | | | | | | | | Lower | Upper |
| Information about drugs tested and used to treat the disease[1] _ Professional degree[2] | Medical staff | 294 | 3.79 | 1.53 | -2.05 | 534 | .03 | -.28 | .13 | -.55 | -.01 |
| | Student | 242 | 4.05 | 1.63 | | | | | | | |

[1] The extent to which respondents believe that information about drugs tested and used to treat the virus was communicated in a coherent manner

[2] Index variable from the professional degrees of respondents. Student: medical student and student at university nursing program, Medical Staff: Senior specialist medical–doctor, Specialist medical–doctor, Resident, Nurse with higher education diploma, Nurse with other studies than higher education

certain variables such as working unit, gender, working unit, living environment, the analysis found no differences between the responses of males and females, of people working in units without COVID– 19 patients and people not working in units with COVID– 19 patients, or in people from the rural and urban area.

In the context of the information about drugs tested and used in the treatment against COVID– 19, the results showed that students believe to a greater extent that such information was communicated in a coherent manner (M = 4.05, SD = 1.63), than the medical staff (M = 3.79, SD = 1.53) (t(534) = -2.05, p<0.05) (Table 7). Hence, one possible explanation for this result would be that, due the experience and knowledge of the medical staff, people who were already working in the healthcare system, such people have greater expectations from authorities when it comes to sending medical information, than medical students.

## (4) What is the perception of medical staff about the role of social media in spreading misinformation about the virus?

The results of the research revealed that respondents were inclined to believe more that social media was a proper environment for spreading fake medical information during the pandemic. By analyzing the information from Table 8, it can be observed that the sum of the responses with negative valences (4.5%) (to an extremely little extent, to a very little extent and to a little extent) is much lower than the sum of the responses with positive valences (89.9%) (to an extremely great extent, to a very great extent, to a great extent). Hence, most participants of the study believe that social media platforms favored the transmission of fake medical news during the pandemic. Furthermore, when trying to find differences in the responses of the participants depending on age, gender, living environment, professional degree or working unit (with COVID– 19 patients or without COVID– 19 patients), we observed that their responses

**Table 8. Perception about the extent to which social media contributed to the spread of medical fake news.**

| | | Frequency | Percent | Valid Percent | Cumulative Percent |
|---|---|---|---|---|---|
| Valid | to an extremely little extent | 2 | .4 | .4 | .4 |
| | to a very little extent | 10 | 1.9 | 1.9 | 2.2 |
| | to a little extent | 12 | 2.2 | 2.2 | 4.5 |
| | nor to a little, neither to a great extent | 30 | 5.6 | 5.6 | 10.1 |
| | to a great extent | 62 | 11.6 | 11.6 | 21.6 |
| | to a very great extent | 88 | 16.4 | 16.4 | 38.1 |
| | to an extremely great extent | 332 | 61.9 | 61.9 | 100.0 |
| | Total | 536 | 100.0 | 100.0 | |

**Table 9. Person correlation between the extent to which social media represents an appropriate environment for sharing official COVID– 19 info and age.**

| | | C1. The extent to which social media represents an appropriate environment for sharing official COVID– 19 info | D2. Age |
|---|---|---|---|
| C1[1]. The extent to which social media represents an appropriate environment for sharing official COVID– 19 info | Pearson Correlation | 1 | -.175** |
| | Sig. (2-tailed) | | .000 |
| | N | 536 | 536 |
| D2[2]. Age | Pearson Correlation | -.175** | 1 |
| | Sig. (2-tailed) | .000 | |
| | N | 536 | 536 |

**. Correlation is significant at the 0.01 level (2-tailed).

[1] C1 –refers to question 1 from the section C of the manuscript (The extent to which social media represents an appropriate environment for sharing official COVID– 19 info), section which refers to Perception about the communication of non- validated treatments

[2] D2—refers to question 2 from the D section of the manuscript (age), which refers to Sociodemographic characteristics of the respondents.

did not differ depending on such variables. Thus, it can be inferred that, regardless of age, gender, living environment, professional degree or working unit, respondents' perception was that social media had a role in spreading fake medical information.

However, even though respondents were of the opinion that social media was an environment in which was sent fake medical information, some of them still believe that social media platforms are appropriate for sending official information about the virus. Thus, considering the results from S3 Table with results to the 4th research question_Table D, the sum of responses with positive valences (40.3%) is almost equal to the sum of responses with negative valences (45.1%) meaning that the opinions of the study participants were divided when it comes to sending official information about the virus on social media.

A factor which showed a weak but statistically significant influence on respondents' opinion about sending COVID– 19 official information on social media was age. Hence, the results of the Pearson correlation (r (534) = -.175, p = 0.000), showed that as age decreases, the extent to which respondents believed that social media is an environment in which official information about the virus should be communicated decreases (Table 9). In other words, younger respondents believed to a greater extent than older respondents that official information should also be communicated on social media. One possible explanation for this results would be that young people gather most of their information from online sources, and they also engage more with social media platforms, and thus it is possible that they would also like to see official and important information on such platforms.

Furthermore, when dividing the study participants in medical staff (doctors, nurses) and students (medical students or students at the university nursing programs), we found that students (M = 4.31, SD = 2.11) believed to a greater extent than the medical staff (M = 3.88, SD = 2.07) that official information about the virus should also be sent on social media (t (534) = -2.36, p< 0.05) (Table 10). Next, when dividing the sample by living environment, participants living in the urban area (M = 4.19, SD = 2.10) were inclined more than those living in the rural area (M = 3.72, SD = 2.05), to believe that official information could also be sent on social media (t (534) = 2.23, p< 0.05) (Table 10).

## (5) What aspects of the professional activity of the medical staff were affected most by the COVID– 19 pandemic?

The findings of our research showed that most respondents stated that the patient–doctor relationship was most affected by the pandemic (38.4%). However, a smaller percent of

**Table 10. Significant t tests for sharing official information on social media professional degree and living environment.**

| | Group | N | Mean | S. D. | t-test for Equality of Means | | | | | | |
|---|---|---|---|---|---|---|---|---|---|---|---|
| | | | | | t | df | p | Mean Difference | Std. Error Difference | CI4 | |
| | | | | | | | | | | Lower | Upper |
| Official information on social media _ Professional degree[1] | Medical staff | 294 | 3.88 | 2.07 | -2.36 | 534 | .01 | -.42 | .18 | -.78 | -.07 |
| | Student | 242 | 4.31 | 2.11 | | | | | | | |
| Official information on social media _living environment | Urban area | 411 | 4.19 | 2.10 | 2.23 | 534 | .02 | .47 | .21 | .05 | .89 |
| | Rural area | 125 | 3.71 | 2.05 | | | | | | | |

[1]Index variable from the professional degrees of respondents. Student: medical student and student at university nursing program, Medical Staff: Senior specialist medical–doctor, Specialist medical–doctor, Resident, Nurse with higher education diploma, Nurse with other studies than higher education

respondents declared that the working schedule was the most affected (26.9%), or the collaboration with their peers (23.9%) (S4 Table with results to the 5th research question_Table E).

Furthermore, taking into account the group of medical staff (doctors, nurses) and the group of students (medical students and students at university nursing program), the results revealed that the most respondents who stated that the patient- doctor relationship was affected most by the pandemic was the group of medical staff (144 compared to 62) (S4 Table with results to the 5th research question_Table F). One possible explanation for this result is that, by being in constant contact with their patients, doctors and nurses were more inclined to perceive that the relation with their patients has deteriorated during the pandemic.

## Discussion

During the COVID– 19 pandemic, one of the major issues people had to face, was the spread of misinformation about the virus, its origins and its treatment. In this regard, we analyzed the perception of medical staff (including doctors, nurses, medical students and students in the university nursing program) about the way medical and non–medical information was communicated during the pandemic. In the context of the so called infodemic [11], and the effects of misinformation on people's trust in doctors, most participants of our study declared that the information about alternative treatments for the virus affected the credibility of health professionals. Hence, from this point of view, our study is in line with previous studies which highlighted the fact that lately, trust in physician decreased [67], and which suggested that social media managed to determine people to trust the personal opinions of other people rather than the opinion of the professionals [61]. Furthermore, since other researchers pointed out that many medical practitioners used social media to express professional opinions that were later found inaccurate [74], and thus they may have contributed to the spread of misinformation [75], we argue that the credibility of physicians might have also been affected by this type of behavior.

An interesting result of our research showed that as the age of medical staff decreases, the extent to which they believe that information about alternative treatments affects doctors' credibility increases. Hence, younger healthcare professionals believed to a greater extent than older healthcare professionals, that information about alternative treatments affected negatively people's trust in doctors. This results might have as possible explanation, the fact that younger people tend to spend more time on social media platforms, and they may have interacted more than older professionals, with misinformation about the virus, this making them more able to be aware of the negative effects of fake news. Moreover, the type of unit in which the respondents worked, was a factor which influenced the opinion of the respondents, our

findings showing that, the medical staff who did not work in unit with COVID -19 patients, believed to a greater extent than those who worked in such units, that information about alternative treatments negatively influenced doctors' credibility. Given this result we argue that is it possible for those professionals who did not interact with COVID -19 patients, and who thus were more distant from the situation, to have a more distorted image regarding the way people's levels of trust in them changed in the context of the pandemic.

Considering the role of social media in spreading misinformation, our study is in line with previous studies which support the idea that such channels favored the communication of fake news during the pandemic [49, 50, 51]. In this regard, regardless of age, professional degree or living environment, most healthcare professionals who participated in our study were of the opinion that social media contributed to the spread of misinformation. However, our study also showed that when it comes to communicating official information on social media, younger respondents (students) believed to a greater extent than older respondents (doctors, nurses), that such channels should be used to send official information about the virus. Taking into account these results, the fact that healthcare professionals acknowledge that social media favors the spread of misinformation, and that many of them still believe they should be used in order to communicate official information, shows that at personal level, professionals were not affected that much by misinformation, them being able to differentiate more easily between real and fake news. In other words, we argue that while people in general were negatively influenced by the fake news they read on social media, as it was shown in previous studies which highlighted that people trusted the information on social networks, they shared un-validated information and had trouble with differentiating real from fake news [57, 79] or that exposure to health misinformation may influence people's intention to engage in certain behaviors [80], healthcare professionals may be less influenced by fake news, due to their knowledge.

Considering the knowledge of medical staff about the type of drugs that had positive effects on treating the virus, the findings of the research showed that the respondents had opinions which were in line with the results found in other studies. Hence, according to the research, most respondents stated that the drug which was known to have positive effects against the virus was Dexamethasone (46.6%), it being followed by Remdesivir (40.5%). Thus, positive effects of Dexamethasone were also highlighted by studies [31, 32], while study [35] showed positive effects of Remdesivir. Moreover, during the period in which we conducted our research, (April–June 2021), among the drugs which were approved for administration against the virus were Remdesivir, Tocilizumab–which was authorized first in June 2021, drugs which were also acknowledged by the respondents of our research. Even more, one of the authors of the article (L.R.) is a doctor and was directly involved in the process of taking care of COVID–19 patients, so the author can confirm that among the drugs which were in trial, or which were approved for administration against COVID-19 were also the drugs which were acknowledged by the respondents of our research.

In the context of medical staff's knowledge about alternative treatments, most respondents declared they had heard about the fact that alcohol can prevent the infection, that warm water drunk every 15 minutes, and the hot air from the hairdryer pointed to the nostrils can help eliminate the virus. From this point of view, our study is in line with a previous study [53], which also described these methods.

When it comes to the respondents' level of satisfaction about the way medical and non–medical information was communicated during the pandemic, generally, the research revealed that most respondents were dissatisfied with the communication process. In the case of communication strategies adopted by authorities, the results showed that most respondents were satisfied with them. However, in the context of sending information about the drugs used to treat the disease, the research showed that younger healthcare professionals were more

satisfied with the communication process than older healthcare professionals. This results might be due to the fact that physicians with more experience have higher expectations from authorities than students.

Another area on which we focused our research was the professional activity of the medical staff during the pandemic. In this regard, our findings revealed that, according to the respondents of our study, the aspect that was mostly affected by the pandemic was the doctor- patient relationship. Hence, our research is in line with other studies [78], which showed that the pandemic affected the way doctors interacted with their patients.

Furthermore, on the basis of the results of our study we argue that not only the process of vaccination created ethical issues, but also the process of communication [81]. Thus, these ethical issues were perceived by the medical staff and they would require a further examination in order to be able to create communication guides which can be regarded as essential instruments not only for the research process of the medical staff and healthcare professionals with management positions, but also for their current medical activity [82, 83].

## Conclusions

During the pandemic, healthcare professionals did not have to deal only with challenges regarding their health and the health of their patients, but also with the problems created by the spread of medical misinformation. According to the main findings of our research, generally, the medical staff (doctors, nurses, medical students, students at university nursing program), believed that information about alternative treatments affected people's trust in doctors, but younger healthcare professionals and those working in units without COVID—19 patients believed to a greater extent than older healthcare professionals and people working in units with COVID– 19 patients that fake news about treatments for the virus affected the credibility of doctors.

Furthermore, regardless of age, age, gender, living environment, professional degree or working unit, the medical staff acknowledged the role of social media in spreading fake news, but when it comes to using social media in order to communicate official information, younger healthcare professionals were more inclined to believe that such networks were appropriate for the communication of official information.

In the context of the drugs used to treat the virus, the results pointed out that the medical staff had knowledge about the drugs known to have positive effects in treating the virus, their perception being in line with previous studies which focused on this matter.

When it comes to the influence of the pandemic on the professional activity of the medical staff, the respondents declared that the aspect which was most affected was the doctor–patient relationship. In this regard, we argue that, by influencing peoples' trust in doctors, the medical fake news spread during the pandemic, implicitly had a role in deteriorating the relation between doctors and their patients.

Therefore, the healthcare professionals were generally dissatisfied with the way medical and non–medical information was communicated during the pandemic, but younger professionals were satisfied than older professionals. Overall, the medical staff believed that fake news managed to undermine doctors' credibility that social media platforms favor the spread of such news, and they had knowledge about the drugs which were known to have positive effects on the virus and about the alternative treatments.

Taking into account the results of the research, the paper has some theoretical and practical implications. From a theoretical point of view, the paper contributes to the literature on the matter of fake news and its influence on the trust of healthcare professionals, a strength of the paper being the fact that it analyzed the opinions of medical staff (doctors, nurses, medical

students and students at university nursing program). From a practical point of view, the paper brings awareness to the phenomenon of fake news regarding medical treatments and the negative influence it has on doctors' credibility. Another practical implication refers to the fact that the paper brings attention to the issue of using social media as a mean to communicate official information, many healthcare professionals, especially the younger ones, stating that such networks could be appropriate for sharing official information. Furthermore, by highlighting that the most affected aspect of the professional activity of doctors was the relationship with their patients, the study also shows that actions need to be taken in order to restore people's trust in doctors and improve the process of communication between them.

## Limitations and future research directions

While our study proved relevant information regarding the perception of healthcare professionals about the way medical and non–medical information was communicated in time of the pandemic, it also has some limitations.

One limitation is represented by the fact that the perception of healthcare professionals was studied only by using quantitative methods. In this regard, a future research should focus on obtaining information from doctors while using qualitative methods too. Next, the study was conducted only on Romanian healthcare professionals, and thus, a future research should take into consideration a comparison between the opinions of professionals from different countries. Another limitation is represented by the fact that we only asked respondents to state the aspect which was most influenced by the pandemic, but we did not asked them to offer detail about other type of challenges encountered. Thus, a future research should focus on analyzing the extent to which aspects of the professional activity of doctors were affected, and on analyzing more deeply the challenges they had to face during the pandemic.

Furthermore, since our research revealed that many respondents believed that social media platforms could be appropriate for sharing official information, we draw attention to a problem that can arise in this context. Since people know that such platforms favor the spread of fake news, if we encourage the use of social media in order to communicate official information, don't we risk to discredit that information as it is possible for people to consider that such information is fake too? We believe that this issue should be taken into account and studied in a future research.

## Supporting information

**S1 Appendix. English version of the questionnaire.**
(DOCX)

**S2 Appendix. Romanian version of the questionnaire.**
(DOCX)

**S1 Table. Results to the 2nd research question.**
(DOCX)

**S2 Table. Results to the 3rd research question.**
(DOCX)

**S3 Table. Results to the 4th research question.**
(DOCX)

**S4 Table. Results to the 5th research question.**
(DOCX)

## Author Contributions

**Conceptualization:** Claudiu Coman, Maria Cristina Bularca, Angela Repanovici.

**Data curation:** Claudiu Coman, Maria Cristina Bularca, Liliana Rogozea.

**Formal analysis:** Claudiu Coman, Maria Cristina Bularca.

**Investigation:** Claudiu Coman, Maria Cristina Bularca, Angela Repanovici, Liliana Rogozea.

**Methodology:** Claudiu Coman, Maria Cristina Bularca.

**Project administration:** Claudiu Coman, Angela Repanovici, Liliana Rogozea.

**Resources:** Maria Cristina Bularca, Angela Repanovici, Liliana Rogozea.

**Supervision:** Claudiu Coman, Angela Repanovici, Liliana Rogozea.

**Writing – original draft:** Claudiu Coman, Maria Cristina Bularca.

**Writing – review & editing:** Claudiu Coman, Maria Cristina Bularca, Angela Repanovici, Liliana Rogozea.

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
