## [Editor Report · Decision Letter 0]

5 Apr 2022

PONE-D-22-09134Challenges in the communication process during the COVID-19 pandemic- a perspective of medical staffPLOS ONE

Dear Dr. Coman,

Thank you for submitting your manuscript to PLOS ONE. Please change the format of the supporting information into .zip, because the .rar format cannot be read. Once this is done we can process your manuscript further.

We look forward to receiving your revised manuscript.

Kind regards,

Markus Ries, MD PhD MHSc FCP

Academic Editor

PLOS ONE Journal requirements:

---

## [Author Response · Author response to Decision Letter 0]

5 Apr 2022

We took into account the comments of the editor and we uploaded again our supporting information in a zip format.

The platform required us to upload a version of the manuscript with track changes but we were not required to make any changes to the manuscript.

The comments of the editor metioned only to upload our supporting information in zip format.

---

## [Decision Letter · Decision Letter 1]

17 May 2022

PONE-D-22-09134R1Challenges in the communication process during the COVID-19 pandemic- a perspective of medical staffPLOS ONE

Dear Dr. Coman,

Thank you for submitting your manuscript to PLOS ONE. After careful consideration, we feel that it has merit but does not fully meet PLOS ONE’s publication criteria as it currently stands. Therefore, we invite you to submit a revised version of the manuscript that addresses the points raised during the review process.

We look forward to receiving your revised manuscript.

Kind regards,

Markus Ries, MD PhD MHSc FCP

Academic Editor

PLOS ONE

Additional Editor Comments (if provided):

Please pay close attention to the reviewers' comments in terms of length, focus, and clarity in the presentation of your study. Some reviewer comments are available in an attachment file, please make sure that you do not miss this. Also, please include an appropriate research checklist that will help you to further improve your manuscript (STROBE https://www.strobe-statement.org/). 

Was your study pre-registered in a study register (e.g., clinicaltrials.gov)? If yes, please state this, if not, please explain why not.

Reviewers' comments:

Reviewer's Responses to Questions

**Comments to the Author**

1. If the authors have adequately addressed your comments raised in a previous round of review and you feel that this manuscript is now acceptable for publication, you may indicate that here to bypass the “Comments to the Author” section, enter your conflict of interest statement in the “Confidential to Editor” section, and submit your "Accept" recommendation.

Reviewer #1: All comments have been addressed

Reviewer #2: (No Response)

2. Is the manuscript technically sound, and do the data support the conclusions?

Reviewer #1: Partly

Reviewer #2: (No Response)

3. Has the statistical analysis been performed appropriately and rigorously? 

Reviewer #1: Yes

Reviewer #2: (No Response)

4. Have the authors made all data underlying the findings in their manuscript fully available?

Reviewer #1: Yes

Reviewer #2: (No Response)

5. Is the manuscript presented in an intelligible fashion and written in standard English?

Reviewer #1: Yes

Reviewer #2: (No Response)

6. Review Comments to the Author

Reviewer #1: the review comments attached. The required modifications can be summarized as following and the authors will find it in details in the attached file:

the authors should review the journal guidelines and abide by it in manuscript preparation.

the introduction section is too long and need to be summarized.

the section titles need to be reviewed and fixed.

the resuklts section include too much tables need to be focusing on the most significant tables and attach the other tables as supplementary tables.

the methods section is missing the research design, sampling method and the calculation of the study sample and the validity and reliability section.

the conclusion section need to be summarized and conclude the main study findings and its significance.

the references are too much need to be filtered and summarized to 30 or 40 refrences maximum.

regards,

Reviewer #2: The study is interesting and shows the point of view of health professionals, misinformation affected trust with the patient. Another fact is that even among professionals, there are different perceptions about the spread of fake news, according to age and occupation. Thus, I suggest adequacy in the title, as it is not expressing exactly what the study observed.

The survey instrument was validated by a sufficient number of professionals; however, I did not find the attached instrument to be evaluated and to verify that the questions supported the statistical data that was generated.

It is important to send supplementary material S1 so that the reviewer can evaluate the work impartially.

The Information on drugs used to treat COVID 19 topic of the Literature review covers the year 2020 and serves to locate the context that health professionals were in at the time of answering the questionnaire, however, there is a lack of information on the drugs that were being recommended by the WHO in the period of application of the questionnaire, which was from April to June 2021.

Contextualizing how the data were in the period when the instrument was applied can directly impact the conclusion: “Healthcare professionals knew about the drugs used in clinical trials”

Minor revisions:

When reading, there are differences in font size/type. E.g. lines 206 and. 534

7. PLOS authors have the option to publish the peer review history of their article (what does this mean?). If published, this will include your full peer review and any attached files.

Reviewer #1: **Yes: **Dr. Sally Mohammed Farghaly

Reviewer #2: No

---

## [Author Response · Author response to Decision Letter 1]

30 Jun 2022

*For a more proper view of our Response to reviewers, we kindly ask you to check the Word document entitled Response to reviewers. 

Dear Sir/Madam

With this cover letter we submit the revised manuscript, initially entitled” Challenges in the communication process during the COVID-19 pandemic- a perspective of medical staff”, and after complying with the suggestions of the reviewers, entitled “Misinformation about medication during the COVID – 19 pandemic: a perspective of medical staff” by Claudiu Coman, Maria Cristina Bularca, Angela Repanovici and Liliana Rogozea for publication in PLOS ONE.

We revised the manuscript according to the suggestions and recommendation made by the reviewers. We would like to thank the reviewers for taking time to review our paper and for providing such useful suggestions. We also thank the academic editor for reviewing our paper. We tried to comply with all the suggestions and recommendations made by the reviewers, and in this letter, we describe the changes we made to the text according to the recommendations of the reviewers. 

Our manuscript needed major revisions. The changes were made while having active the “Track changes” function from Microsoft Word and the lines where the text was changed can be best viewed while having active the “All markup” option. Moreover, in order for our changes to be best seen, we will also provide in this cover letter, the lines from the revised manuscript with the “Track changes” function, and “All markup” option active. With regards to our response to Reviewer 1, the reviewer made a series of suggestions directly in the PDF version of our initial manuscript, but also provided a summary of those suggestions in the e-mail which was sent by the journal to the corresponding author. In this regard, we responded first to the comments highlighted in the summary from the e-mail, and then we responded to each point made by Reviewer 1 in the PDF version of our initial manuscript. Next, we responded to each point raised by Reviewer 2.

Our response to Reviewer 1: 

We firstly thank the reviewer for taking time to review our manuscript and provide suggestions in order to improve it. We addressed all the suggestions made by the reviewer. When we describe how the text was changed, we also provide the lines where the text can be found in the revised manuscript with the option “Track changes” active. In this way, the changes can be viewed completely (the text we deleted, and the text we inserted). Next, we will firstly describe our answers to the comments which were summarized in the e-mail received by the corresponding author, and then we will present our responses to the comments made by the reviewer in the PDF version of our manuscript.

Reviewer 1 comments- as summarized in the email received by the corresponding author 

Reviewer 1 point 1: the review comments attached. The required modifications can be summarized as following and the authors will find it in details in the attached file: the authors should review the journal guidelines and abide by it in manuscript preparation.

Response 1: We are grateful to the reviewer for the suggestion. We reviewed the guidelines of PLOS ONE journal again and we made sure our manuscript is prepared in accordance to the author guidelines which can be found on the journal’s official website. We also checked the pdf files entitled “Download sample title, author list, and affiliation page” and “Download sample manuscript body”, in order to make sure our manuscript is correctly formatted. Thus, we looked again at the guidelines for the sections which have to be included in the manuscript, the font and sizes for headings, table captions, referencing rules, etc., and we made sure our manuscript respects the guidelines of the journal.

Reviewer 1 point 2: the introduction section is too long and need to be summarized.

Response 2: We thank the reviewer for the useful suggestion. In order to comply with it, we tried to summarize our introduction. Thus, we would like to mention that we also took into account the comments the reviewer made in the pdf version of the manuscript. In this regard, there the reviewer recommended us to rephrase the first paragraph of our paper because the paragraph was not about the communication process: “the introductory paragraph is not related to communication process”. We rephrased the paragraph and we added information in which we highlighted the fact that the COVID – 19 pandemic negatively influenced the communication process. The changes we made, the text deleted, added or rephrased can be best seen while having active the “Track changes” function and the “All markup” option provided by Microsoft Word. Thus, in the Introduction section of the paragraph, page 4 of the manuscript, lines 71-77, we made changes to the text, and the new introductory paragraph also addresses the subject of communication: 

“The COVID 19 pandemic generated multiple changes in the way today’s society members carry out their daily activities. One of the processes which was mostly affected by the pandemic was the communication process between institutions and the public, as well as between individuals. In this regard, from this perspective, while many domains were affected by the spread of the virus, such as the educational system or the cultural sector, the health sector was the one that faced the most challenges [1].”

Next, in the pdf version of our manuscript, the reviewer suggested that the details we gave regarding the virus could be summarized in one paragraph: “the history of covid-19 can be summarized in a single paragraph”. In order to comply with the request, in the Introduction section, at page 4 of the manuscript, we summarized the text indicated by the reviewer.

The text the reviewer suggested us to summarize:

“Caused by severe acute respiratory syndrome coronavirus 2 (SARS-CoV-2) [2], the disease was firstly detected in December 2019, in Wuhan, China [3], and it fastly spread all over the world. The World Health Organization was informed about a pneumonia outbreak in Wuhan on December 31 2019, the number of cases continued to increase, and on March 11 2020 the World Health Organization characterized COVID 19 as a pandemic [4]. Being highly contagious, the virus affected a large number of people, and as of November 27 over 61 million cases were reported [5]. Even though many companies and institutions are struggling to develop a vaccine, Pfizer, Gamaleya Research Institute, University of Oxford, and a preliminary analysis of the vaccine proposed by Pfizer showed that the vaccine is able to prevent more than 90% of people from getting infected with COVID 19 [6], so far no vaccine was approved as a general and universal vaccine against COVID 19 [7]. Ever since the pandemic was declared, finding the right treatment for the virus has become a priority for researchers and doctors from all over the world. In this regard, large number of trials started to be conducted, and in order to find an efficient drug treatment against the virus, one method that was adopted was testing and administrating to patients, drugs that were previously used for curing other viruses [8]. Thus, on March 20 2020, The World Health Organization launched the SOLIDARITY clinical trial, a trial that monitored the effects on patients infected with COVID 19, of specific drugs that proven to be effective in the treatment of other diseases: remdesivir, interferon beta, chloroquine and hydroxychloroquine -previously used for Malaria, as well as drugs used on HIV patients: lopinavir and ritonavir [9]. However, according to the interim results published on October 15 2020 by WHO, even though those drugs were taught to have positive effects on treating COVID 19, they had little influence or no influence at all on mortality in general, on the need and initiation of ventilation and on the recovery process [10].”

The way we summarized the text can be seen at lines 102-114- in the revised version of our manuscript (The full change, the text deleted and the text summarizes is visible at lines 78-114).

The text we summarized (lines 102 -114 with the “Track changes” and “All Markup” option active:

“Caused by severe acute respiratory syndrome coronavirus 2 [2], the disease was firstly detected in December 2019, in Wuhan, China [3]. Due to the evolution of the virus, the World Health Organization declared the pandemic in March 2020 [4], and as of November 27 over 61 million cases were reported [5]. In this regard, although several companies are struggling to develop a vaccine, and some of the proposed vaccines showed promising results [6], so far no vaccine was approved in order to be administrated to the entire population [7]. Ever since the pandemic was declared, many companies started to be preoccupied with finding a treatment, and one method used that was adopted was administrating to patients, drugs that were previously used for curing other viruses [8]. Thus, one of the most well - known trials started was the SOLIDARITY trial, which focused on using various drugs including chloroquine and hydroxychloroquine, lopinavir or ritonavir [9]. However, even if those drugs were taught to have positive effects on treating the virus, they did not have a significant influence on preventing mortality in general [10]”.

Next, in order to reduce the information written in the Introduction section, as the reviewer suggested, we also deleted the last paragraph of the Introduction section, paragraph in which we provided details about the concepts that we addressed next in the Literature review section. Thus, at lines 118 – 122 in the revised manuscript with “Track changes” and “All Markup” option active, we deleted the following text:

“Hence, considering the purpose of our paper and the research questions, we believed it was necessary to analyze the literature on the drugs used to treat COVID – 19, on the role of social media platforms in spreading fake information about the virus and potential treatments, and on the way the pandemic influenced the credibility of doctors and their relationship with their patients.”

Reviewer 1 point 3: the section titles need to be reviewed and fixed. 

Response 3: We thank the reviewer for the useful suggestion. We checked again the author guidelines provided by the journal on its official website, regarding sections of the manuscript. In this regard, we corrected the section which was entitled “Methods and materials” in the initial version of our manuscript, with the correct form, which is “Materials and methods”. The change can be seen in the revised manuscript at page 17, line 364, while having active the “Track changes” and “All markup” options from Microsoft Word. We reviewed all of our section titles and made sure they are correct.

Reviewer 1 point 4: the resuklts section include too much tables need to be focusing on the most significant tables and attach the other tablesas supplementary tables.

Response 4: We are grateful to the reviewer for such useful suggestion. We addressed the suggestion, we looked at the tables included in the Results section and we integrated in the section only the most significant tables. The other tables were deleted from the text and added to supplementary information. Thus, we created Word documents with supplementary information for each of our research questions. In this regard in S3_Tables with results to the 1st research question we included Table 2 ; in S4_Tables with results to the 2nd research question we included Table 5 and Table 6; in S5_Tables with results to the 3rd research question we included Table 7 and Table 9; in S6_Tables with results to the 4th research question we included Table 11 and Table 12; in S7_Tables with results to the 5th research question we included Table 15 and Table 16. 

Reviewer 1 point 5: the methods section is missing the research design, sampling method and the calculation of the study sample and the validity and reliability section.

Response 5: We are very grateful to the reviewer for suggesting us to improve the methods section of our paper. With regards to the research design section, we added this section to our manuscript and we explained in detail the research design. Even more, we deleted some information from the Sampling and data collection procedures and we added it to the research design section because it was more suitable there. In this regard, at pages 17-18 of the manuscript, between lines 365- 385 can be found the Research design section of our paper, which comprises the following text:

“The present study was conducted on Romanian healthcare professionals including doctors, nurses and medical students. The method used is quantitative. The questionnaire was administrated online, the data was collected through the help of Google forms, and was disseminated on groups of healthcare professionals and students on platforms such as Facebook and WhatsApp, during the period April 2021– June 2021. The data we collected was firstly exported to Microsoft Excel, and then it was analyzed with IBM Statistical Package for the Social Sciences, version 20. The respondents were informed about the purpose of the study, about the fact that they were allowed to withdraw at any time, and they were asked to give their consent for participating in the study. The average time needed to complete the questionnaire was 15 minutes. Considering the validity of our research, we took into account the theoretical information from the literature regarding the development of a questionnaire. Our team of researchers together with health specialists have configured the dimensions, and operationalized the concepts in accordance with the theoretical approaches identified at the current stage of the research. Even more, we pre-tested the questionnaire before disseminating in order to guarantee the validity of the instrument. Thus, the questionnaire was completed by 50 respondents in the pre-testing stage. Considering the reliability of the research, we used split half reliability method. We split our sample in half, and we checked the variables in from our sub-samples in order to see if the variables provided convergent results. The convergent results we obtained by applying the split half method showed that we obtained a high fidelity measurement.

In order to create the research design section and to also improve the way our paper is structured, we made changes to the section “Sampling and data collection procedures”. In this regard, we deleted some text and we reformulated some phrases. The section comprises the following text, which can be found at pages 17-18 of the revised manuscript with “Track changes” and “All markup” option active, lines 419-427:

“In order to conduct the research we used a quantitative method while having as an instrument a questionnaire. The responses were collected online, with the help of Google forms, and the questionnaire was self – administrated. The research received approval from The Council of the Faculty of Sociology and Communication, approval request Nr.378/30.03.2021. Taking into account the sampling method and the calculation of the study sample, we used random, probabilistic sampling method. We took into consideration specialists, physicians, and medical students from Brasov, and we applied the snowballing method in order to disseminate the questionnaire. The sample of our study comprises 536 respondents, and included doctors, nurses as well as medical students from Romania.”

With regards to the sampling method, we would like to thank the reviewer for pointing out that we should give more information about the sampling procedure. Even though in the initial version of our manuscript we described the sample of our research, how the questionnaire was distributed and to whom, we added more specific information about the sampling method. Hence, at page 20 of the manuscript, lines 423 - 426, we explained that we used a random, probabilistic sampling method: 

“Taking into account the sampling method and the calculation of the study sample, we used random, probabilistic sampling method. We took into consideration specialists, physicians and medical students from Brasov, and we applied the snowballing method in order to disseminate the questionnaire.” 

Reviewer 1 point 6: the conclusion section need to be summarized and conclude the main study findings and its significance. 

Response 6: We are grateful to the reviewer for the suggestion. In order to comply with it we tried to summarize our Conclusions section, to highlight again the main findings of the research and the significance of our study. In this regard, the text which was written in Conclusions in the initial version of our manuscript was improved. In this regard, we deleted some of the redundant information which was written in this section. The information we deleted:

“In this regard, besides fighting the pandemic, physician also had to fight the so called infodemic. Fake news spread on social media about various alternative treatments for the virus and the opinions of certain professionals about treatment methods which later proven to be inaccurate negatively influenced the credibility of doctors.” (Lines 789-792)

“This results can suggest that while professionals were aware of the role of social media in spreading medical misinformation and in affecting trust in doctors, due to their knowledge, at personal level they were less affected by that type of information, many of them believing that social media should also be used for sending official information” (lines 803-807)

“Moreover, the medical staff was aware of the alternative treatments which were promoted on social media, the method of drinking alcohol in order to prevent the infection being the method that most of the respondents have heard about” (lines 811-813).

“Hence, on the basis of the findings and implications of the study, we further discuss limitations and future research directions.” (Lines 838-839).

Next, we took into account the recommendation of the reviewer and we started the section by presenting the main findings of our research. Since we had several research questions, we presented our main findings in relation to those research questions. Next, the reviewer recommended us to explain the significance of our study. Thus, in the paper we had already written the theoretical and practical implication of our paper. In this regard, we did not delete the implications because we consider that the implications emphasize why the study conducted is important and how it can be further taken into consideration. Next, we did not delete the limitations and future research directions either, because we considered necessary to highlight how and why our study has limitations but also how it could be further developed or extended. 

Reviewer 1 point 7: the references are too much need to be filtered and summarized to 30 or 40 refrences maximum. Regards

Response 7: We are very grateful to the reviewer for this recommendation and we appreciated the interest in improving our paper. However, when we started to write the article, we wanted to make sure our paper will be well documented and that it will address all the theoretical concepts and aspects needed. In this regard, we made a thorough research and literature review on the medication used in order to treat the virus, on the way social media contributed to the spread of misinformation about the virus, and on the way trust in doctors and the doctor- patient relation was affected during the pandemic. Thus, we read many research paper because we wanted for our paper to provide an overall view on the subject addressed. In this regard, we consider that all the references we used are relevant for the subject approached and for the research that we conducted, and therefore we could not delete more than half of them. In other words, through the references cited we support and sustain our arguments, we show how other researchers approached similar matters and thus we could not delete more than half of our references because we considered that by deleting them we could no longer have a strong and well consolidated theoretical background and we could not properly explain how we wanted to address the matted of medical misinformation and its effects from the perspective of medical staff. Even more, the journal does not have a limitation regarding the length of the article or the number of references: “Manuscripts can be any length. There are no restrictions on word count, number of figures, or amount of supporting information”. In addition, we have seen articles which addressed subjects related to health and the COVID – 19 pandemic, and which were published in PLOS ONE, that have more than 40 references. For example, one article entitled “Severity of infection with the SARS- CoV -2 B1.1.7 lineage among hospitalized COVID – 19 patients in Belgium” (https://journals.plos.org/plosone/article?id=10.1371/journal.pone.0269138), has 76 references, and another article, entitled “The coronavirus disease 2019 (COVID -19) vaccination psychological antecedent assessment using the ARABIC 5c validated tool: An online survey in 13 Arab countries” (https://journals.plos.org/plosone/article?id=10.1371/journal.pone.0260321) has 71 references.

Reviewer 1 comments- as pointed by the reviewer in the PDF version of our manuscript

Reviewer 1 point 1: A perspective of medical staff

Response 1: We thank the reviewer for the suggestion. We put “:” instead of “-“in our title, before the phrase “a perspective of medical staff”. The change can be seen at line 2 of the revised manuscript. 

Reviewer 1 point 2: the abstract need to be summarized to 250 to 300 words by the main important information in each part ....it is recommended to avoid long paragraphs and to paraphraze and summarize the ideas in short paragraphs.

Response 2: We are grateful to the reviewer for the recommendation. In order to comply with it we summarized our abstract to 219 words. In this regard, we deleted the text which was written in the Abstract section, and instead, at page 3 of the revised manuscript with “Track changes” and “All markup” option on, at lines 50 –68 we inserted the following text:

“Background. Healthcare professionals had to face numerous challenges during the pandemic, their professional activity being influenced not only by the virus, but also by the spread of medical misinformation. In this regard, we aimed to analyze, from the perspective of medical staff, the way medical and non - medical information about the virus was communicated during the pandemic in order to raise awareness about the way misinformation affected the medical staff. 

Methods and findings. The study was conducted on Romanian healthcare professionals. They were asked to answer to a questionnaire and the sample of the research includes 536 respondents. The findings revealed that most respondents stated that information about alternative treatments against the virus affected the credibility of health professionals, and that younger professionals believed to a greater extent that trust in doctors was affected. The research also showed that respondents were well informed about the drugs used in clinical trials in order to treat the virus.

Conclusions. Healthcare professionals declared that the spread of misinformation regarding alternative treatments, affected their credibility and the relationship with their patients. Healthcare professionals had knowledge about the drugs used in clinical trials, and they acknowledged the role of social media in spreading medical misinformation. However, younger professionals also believed that social media could be used to share official information about the virus.”

Reviewer 1 point 3: the introductory paragraph is not related to communication process.

Response 3: We thank the reviewer for pointing this out. We explained how we addressed this point above in this Cover letter, in point 2 raised by the reviewer in the summary which was written in the e-mail sent to the corresponding author. However, we will present again the way we changed the introductory paragraph in order for it to be related to communication process. In this regards, in the Introduction section of the paragraph, page 4 of the manuscript with “Track changes” and “All markup active”, lines 71-77, we made changes to the text, and the new introductory paragraph also addresses the subject of communication: 

“The COVID 19 pandemic generated multiple changes in the way today’s society members carry out their daily activities. One of the processes which was mostly affected by the pandemic was the communication process between institutions and the public, as well as between individuals. In this regard, from this perspective, while many domains were affected by the spread of the virus, such as the educational system or the cultural sector, the health sector was the one that faced the most challenges [1].”

Reviewer 1 point 4: the history of covid-19 can be summarized in a single paragraph.

Response 4: We are very grateful to the reviewer for the recommendation. We tried to comply with it and we summarized the history of COVID -19. Earlier in this cover letter we explained how we addressed this point because the reviewer also mentioned it in the summary which was written in the e-mail sent to the corresponding author. In this regard, we summarized the indicated text, and at page 5 of the manuscript with “Track changes” and “All markup” option active, lines 102- 114 we added the following text:

“Caused by severe acute respiratory syndrome coronavirus 2 [2], the disease was firstly detected in December 2019, in Wuhan, China [3]. Due to the evolution of the virus, the World Health Organization declared the pandemic in March 2020 [4], and as of November 27 over 61 million cases were reported [5]. In this regard, although several companies are struggling to develop a vaccine, and some of the proposed vaccines showed promising results [6], so far no vaccine was approved in order to be administrated to the entire population [7]. Ever since the pandemic was declared, many companies started to be preoccupied with finding a treatment, and one method used that was adopted was administrating to patients, drugs that were previously used for curing other viruses [8]. Thus, one of the most well - known trials started was the SOLIDARITY trial, which focused on using various drugs including chloroquine and hydroxychloroquine, lopinavir or ritonavir [9]. However, even if those drugs were taught to have positive effects on treating the virus, they did not have a significant influence on preventing mortality in general [10]”.

Reviewer 1 point 5: the stydy aim is to assess the perception and this other aim is not included as an intervention, so it is better to rephrased as to recommend future researches or interventions to raise......

Response 5: We thank the reviewer for the useful suggestion. We tried our best in addressing the recommendation. In this regard, we rephrased the part of the purpose indicated by the reviewer. In other words, the reviewer suggested us to rephrase the last part of our purpose, to rephrase the expression “in order to raise awareness about the way misinformation affected medical staff”. Hence, at page 6 of the manuscript with “Track changes” and “All Markup” option active, lines 129 –133 we rephrased the purpose and added the following text:

“The purpose of the paper is to analyze, from the perspective of medical staff, the way medical and non - medical information about the virus was communicated during the pandemic to encourage the development of future research or interventions in order to raise awareness about the way misinformation affected medical staff.”

Due to the suggestion of the reviewer, we had to change the way we described the purpose of our paper in other sections of our manuscript too. Thus, the purpose of the paper was changed in the way recommended by the reviewer, also at lines: 52 -55 (in the Abstract section).

Reviewer 1 point 6: please to consider the restructuring of the manuscript as per the journal guidelines and the title of each section. Also, the literature review section is very long and it should be fixed to bo not more than 2 to 2 and half pages summarizing the main ideas.

Response 6: We are very grateful to the reviewer for suggesting us to check again the guidelines of the journal. As we previously explained in this Cover letter, (due to the fact that the same point was also highlighted by the reviewer in the summary which was written in the e-mail sent to the corresponding author), we checked again the guidelines and made sure our manuscript is formatted according to the guidelines. We also checked again the titles of the section which should be included in the manuscript, and at page 17 of the revised manuscript with “Track changes” and “All markup” option active, line 364 we changed “Methods and materials” to “Materials and methods”. 

With regards to summarizing our Literature review and deleting references from our paper, we present again the explanation we gave earlier in the Cover letter, at point 7 made by the reviewer in the e-mail sent to the corresponding author:

We are very grateful to the reviewer for this recommendation and we appreciated the interest in improving our paper. However, when we started to write the article, we wanted to make sure our paper will be well documented and that it will address all the theoretical concepts and aspects needed. In this regard, we made a thorough research and literature review on the medication used in order to treat the virus, on the way social media contributed to the spread of misinformation about the virus, and on the way trust in doctors and the doctor- patient relation was affected during the pandemic. Thus, we read many research paper because we wanted for our paper to provide an overall view on the subject addressed. In this regard, we consider that all the references we used are relevant for the subject approached and for the research that we conducted, and therefore we could not delete more than half of them. In other words, through the references cited we support and sustain our arguments, we show how other researchers approached similar matters and thus we could not delete more than half of our references because we considered that by deleting them we could no longer have a strong and well consolidated theoretical background and we could not properly explain how we wanted to address the matted of medical misinformation and its effects from the perspective of medical staff. Even more, the journal does not have a limitation regarding the length of the article or the number of references: “Manuscripts can be any length. There are no restrictions on word count, number of figures, or amount of supporting information”. In addition, we have seen articles which addressed subjects related to health and the COVID – 19 pandemic, and which were published in PLOS ONE, that have more than 40 references. For example, one article entitled “Severity of infection with the SARS- CoV -2 B1.1.7 lineage among hospitalized COVID – 19 patients in Belgium” (https://journals.plos.org/plosone/article?id=10.1371/journal.pone.0269138), has 76 references, and another article, entitled “The coronavirus disease 2019 (COVID -19) vaccination psychological antecedent assessment using the ARABIC 5c validated tool: An online survey in 13 Arab countries” (https://journals.plos.org/plosone/article?id=10.1371/journal.pone.0260321) has 71 references.

Reviewer 1 point 7: Research Design (please to review examples of the journal manuscript preparation)

Response 7: We thank the reviewer for pointing out that we should described more thoroughly the Research design of our paper. We explained how we addressed this suggestion earlier in this Cover letter, because the reviewer highlighted the suggestion in the summary from the e-mail sent to the corresponding author too. However, we will present again the way we complied with the suggestion. We did review examples of the journal manuscript preparation, and after we had done so, we deleted some text from the section “Sampling and data collection procedures” and moved it to the new section created. In this regard, at pages 17-18 of the revised manuscript with “Track changes” and “All markup” option active, lines 365-385, we inserted a sub-section entitled “Research design” which comprises the following text:

“The present study was conducted on Romanian healthcare professionals including doctors, nurses and medical students. The method used is quantitative. The questionnaire was administrated online, the data was collected through the help of Google forms, and was disseminated on groups of healthcare professionals and students on platforms such as Facebook and WhatsApp, during the period April 2021– June 2021. The data we collected was firstly exported to Microsoft Excel, and then it was analyzed with IBM Statistical Package for the Social Sciences, version 20. The respondents were informed about the purpose of the study, about the fact that they were allowed to withdraw at any time, and they were asked to give their consent for participating in the study. The average time needed to complete the questionnaire was 15 minutes. Considering the validity of our research, we took into account the theoretical information from the literature regarding the development of a questionnaire. Our team of researchers together with health specialists have configured the dimensions, and operationalized the concepts in accordance with the theoretical approaches identified at the current stage of the research. Even more, we pre-tested the questionnaire before disseminating in order to guarantee the validity of the instrument. Thus, the questionnaire was completed by 50 respondents in the pre-testing stage. Considering the reliability of the research, we used split half reliability method. We split our sample in half, and we checked the variables in from our sub-samples in order to see if the variables provided similar results. The convergent results we obtained by applying the split half method showed that we obtained a high fidelity measurement.”

Reviewer 1 point 8: methods and data (please to review the journal authors guideline).Also the reserch design is missed, please to clarify the research design used.

Response 8: We thank the reviewer for the suggestion. We reviewed again the journal author guidelines. Also, we added a research design section and the text contained in the section can be found at lines 365-385 of the manuscript with the “Track changes” and “All markup” option active.

Reviewer 1 point 9: start new sentence (line 333) in the PDF version of our manuscript

Response 9: We thank the reviewer for the recommendation. We complied with it and we started a new sentence, at page 17 of the manuscript with “Track changes” and “All markup” option active, lines 372 we deleted the words “At the beginning of the questionnaire”, and we started a new sentence with “The respondents were informed…”. 

Reviewer 1 point 10: Also this section should not include the data interpretation or analysis. it should include only description.

Response 10: The reviewer referred to the “Sample and data collection procedure” section. We are grateful to the reviewer for the suggestion and in order to comply with it we made some changes to the text which was written in this section. In this regard, the data interpretation and analysis was removed from the section, and was moved to the “Results” section of our paper. The deleted text together with the table can be seen at lines 427 –444 of the revised manuscript with “Track changes” and “All markup” option active. The text we inserted in the “Results” section can be seen at lines 486-501 of the manuscript:

“Out of the 536 respondents, 460 (85.8%) were female and 76 (14.2%) were male. A total of 411 respondents live in the urban area (76.7%), while 125 (23.3%) live in the rural area. Most respondents (286, 53.4%) are between 18 and 35 years of age, 142 respondents (26.5%) are between 36 and 50 years of age, 102 respondents (19.0%) are between 51 and 65 years of age, and 6 of them (1.1) are over 65 years of age. When it comes to the professional degree of the respondents, most of them are students at a university nursing program (122, 22.8%), and medical students (120, 22.4%). However, a total of 102 respondents (19.0%) are senior specialists medical – doctors, and 70 (13.1%) are nurses who have a higher education diploma. When it comes to the respondents field of specialization, most of them (70.5%) operate in the field of general medicine, while others are family doctors (10.4%), pediatricians (3%), dentists or oncologists (1.9%), surgeons of doctors who are specialized in internal medicine (1.5%), or infectious disease doctors, radiologists or cardiologists (1.1%). Furthermore, most of the respondents (77.2%) stated that they did not work a unit with COVID – 19 patients while few of them (22.8%) stated that they worked in such a unit at the time the research was conducted. Thus, all the characteristics of the sample are presented in Table 1.

Table 1. Sample characteristics (n = 536).

 Category Count Percentage

Gender Female 460 88.8%

 Male 76 14.2%

Living environment Urban 411 76.7%

 Rural 125 23.3%

Age 18-35 years old 286 53.4%

 36-50 years old 142 26.5%

 51 -65 years old 102 19.0%

 Over 65 years old 6 1.1%

Professional degree Senior specialist medical - doctor 102 19.0%

 Specialist medical - doctor 46 8.6%

 Resident 28 5.2%

 Nurse with higher education diploma 70 13.1%

 Nurse with other studies than higher education 48 9.0%

 Medical student 120 22.4%

 Student at university nursing program

 122 22.8%

Field of specialization General medicine 378 70.5%

 Family doctor 56 10.4%

 Pediatrics 16 3%

 Stomatology 10 1.9%

 Oncology 10 1.9%

 Surgery 8 1.5%

 Internal medicine 8 1.5%

 Virology/ infectious disease doctor 6 1.1%

 Cardiology 6 1.1%

 Radiology 6 1.1%

 Other 32 6%

Works in a unit with COVID – 19 patients Yes 122 22.8%

 No 414 77.2%

 ”

Reviewer 1 point 11: please to explain how you calculated the sample size and the type of sampling that you used.

Response 11: We thank the reviewer for the suggestion. We offered an explanation for this point, which was also mentioned by the reviewer in the summary provided in the e-mail sent to the corresponding author. However, we will present again the explanation, which can be found at lines 413-416 of the manuscript with “Track changes” and “All markup” option active:

“Taking into account the sampling method and the calculation of the study sample, we used random, probabilistic sampling method. We took into consideration specialists, physicians and medical students from Brasov, and we applied the snowballing method in order to disseminate the questionnaire.” 

Reviewer 1 point 12: this section should be trasfered before data presentation and analysis with the methods part before data analysis

Response 12: The reviewer was referring to “The research instrument” section. We thank the reviewer for the suggestion. Since the section was already written before the “Data analysis” section, we moved the section before “Sampling and data collection procedures”. The deleted text can be seen at lines 446-465 in the revised the manuscript with “Track changes” and “All markup” option active. The section was moved and so, the following text can be found in the revised manuscript at lines 387-406:

“In order to conduct the research we used a quantitative method while having a questionnaire as an instrument. In this regard, we developed a questionnaire which comprises four sections: A. Influence of the pandemic on the professional activity of medical staff (items A1 to A4), B. Perception about the authorities’ communication process (items B1 to B11), C. Perception about the communication of non- validated treatments (items C1 to C20), and D. Sociodemographic questions (items D1 – D9), such as: gender, age, living environment, professional degree, field of specialization. The sociodemographic questions were used in order to identify different or similar attitudes between specific groups. The questionnaire can be found in “S1.Appendix English version of the questionnaire”, and in “S2. Appendix Romanian version of the questionnaire.” Before disseminating the questionnaire, the instrument was tested on 30 doctors who work in the field of cardiology and general medicine. The respondents understood clearly the questions and did not report any issue in the process of answering them. Hence, the questionnaire comprises close ended and open ended questions (Items A1, A4, B3, B11, C19, C20, D2, D5, D6,) dihotomic questions as well as questions whose answers were measured on a 7 point Likert scale. For example, item A2 measured the extent to which the respondents considered that the pandemic influenced the way they carried out their professional activity (1- “to an extremely little extent, 7 “to an extremely great extent”), or item B2 measure the respondents’ level of agreement with statements regarding the way authorities communicated during the pandemic (1 – “strongly disagree, 7-“strongly agree”).”

Reviewer 1 point 13: the validity and reliabity section is missed , please to discuss it clearly Response 13: We thank the reviewer for the recommendation. In order to address the recommendation, we inserted into our manuscript information about the validity and reliability of our research in the “Research design” section. In this regard, at page 18 of the manuscript with “Track changes” and “All markup” option active, lines 376 – 385, we inserted the following explanation:

“Considering the validity of our research, we took into account the theoretical information from the literature regarding the development of a questionnaire. Our team of researchers together with health specialists have configured the dimensions, and operationalized the concepts in accordance with the theoretical approaches identified at the current stage of the research. Even more, we pre-tested the questionnaire before disseminating in order to guarantee the validity of the instrument. Thus, the questionnaire was completed by 50 respondents in the pre-testing stage. 

Considering the reliability of the research, we used split half reliability method. We split our sample in half, and we checked the variables in from our sub-samples in order to see if the variables provided similar results. The convergent results we obtained by applying the split half method showed that we obtained a high fidelity measurement.”

Reviewer 1 point 14: you have two tables number by number 1 two times. please to review the tables numbering and indexing in the maneuscript.

Response 14: We are very grateful to the reviewer for pointing this out. We checked again all the numbers of the tables and corrected all the mistakes. Now in the revised manuscript, all the tables are correctly numbered.

Reviewer 1 point 15: these codes need to be interpretted ( to give its full interpretaion under each table)

Response 15: We thank the reviewer for the suggestion. The reviewer was referring to the numbers of the questions which appear in the tables with correlations and t tests. Those numbers represent the number of the questions from the questionnaires which were included in the t tests or in the correlations. In other words, the numbers refer to the variables used in order to make the tests and the correlations. For example, in Table 3, C14 means, the question 14 from the questionnaire, which belongs to section C. Section C refers to Perception about the communication of non- validated treatments. So, under each table from our manuscript (including the tables which we put in supplementary information) we added an explanation of the codes (numbers). 

We would like to mention that the numbers of our tables changed, because in the initial manuscript we had two tables numbered 1, so now we corrected the mistake. Thus, we further present the explanation we gave in the revised manuscript with “Track changes” and “All markup” option active, under each table:

Table 3 (which was table 2 in the initial manuscript). The following explanation was added under the table: “1 1 C14 – refers to the question 14 from the section C of the manuscript (The extent to which information about alternative treatments affected trust in physicians), section which refers to Perception about the communication of non- validated treatments; 2D2 - refers to question 2 from the D section of the manuscript (age), which refers to Sociodemographic characteristics of the respondents

Table 8 (which was Table 7 in the initial manuscript). The following explanation was added under the table “1 B10- refers to the question 10 from the section B of the manuscript (Satisfaction with the way information about drugs used to treat the virus was communicated) section which refers to Perception about the authorities’ communication process; 2D2 - refers to question 2 from the D section of the manuscript (age), which refers to Sociodemographic characteristics of the respondents.”

Table 13 (which was Table 12 in the initial manuscript). The following explanation was added under the table “1 C1 – refers to question 1 from the section C of the manuscript (The extent to which social media represents an appropriate environment for sharing official COVID – 19 info), section which refers to Perception about the communication of non- validated treatments; 2D2 - refers to question 2 from the D section of the manuscript (age), which refers to Sociodemographic characteristics of the respondents.

Table 16 (which was Table 15 in the initial manuscript and which is in Supplementary information - S7 Tables with results to the 5th research question). The following explanation was added under the table “2A3 – refers to question 3 from the section A of the manuscript (Main aspect of professional life influenced by the pandemic), section which refers to Influence of the pandemic on the professional activity of medical staff; The explanation for 1 professional degree was already written under the table in the initial version of our manuscript.

Reviewer 1 point 16: the variables need to be clear on the table

Response 16: We thank the reviewer for pointing this out. The reviewer was referring to the variables from the table which had the number 3 in the initial version of our manuscript. The table now has the number 4, because we corrected the way we numbered the tables. Hence, in order to be clear which the variables in the table are, we put the word “variables” in front of the variables which were tested. The changes to the table can be seen in the revised version of our manuscript with “Track changes” and “All markup” option active at page 29:

“Table 4. Significant t-test results: comparisons between variables

 t-test for Equality of Means

 Group N Mean S. D. t df p Mean Difference Std. Error Difference CI4

 Lower Upper

Variables: Information about alternative treatments _ Professional degree1 Medical staff 294 5.33 1.54 -2.04 534 .04 -.27 .13 -.52 -.01

 Student 242 5.60 1.49 

Variables: Information about alternative treatments _working unit Unit with COVID -19 patients 122 5.19 1.61 -2.13 534 .03 -.33 .15 -.64 -.02

 Unit without COVID 19 patients 414 5.53 1.49 

Variables: Information about alternative treatments _gender Male 76 5.10 1.70 -2.16 534 .03 -.40 .18 -.77 -.03

 Female 460 5.51 1.48 

1Index variable from the professional degrees of respondents. Student: medical student and student at university nursing program, Medical Staff: Senior specialist medical – doctor, Specialist medical – doctor, Resident, Nurse with higher education diploma, Nurse with other studies than higher education”

Reviewer 1 point 17: there keys need to be written in full interpretaion under each table.

Response 17: We thank the reviewer for the recommendation. We complied with it, and as we explained at one of the previous points of the reviewer, the keys (or codes) refer to the number of the question from the questionnaire, and the letter refers to the section of the questionnaire. Hence, the reviewer referred to the table which had the number 7 in the initial version of our manuscript. The table has the number 8 in the revised version of our manuscript with “Track changes” and “All markup” option active, because we corrected the way we numbered the tables. Under table 8, at page 33 of the manuscript we added the following explanation:

“1 B10- refers to the question 10 from the section B of the manuscript (Satisfaction with the way information about drugs used to treat the virus was communicated) section which refers to Perception about the authorities’ communication process; 2D2 - refers to question 2 from the D section of the manuscript (age), which refers to Sociodemographic characteristics of the respondents.”

Reviewer 1 point 18: the tables are too much, please to focus on the highly significant tables and add the others as a supplementary tables. it is recommended to reduce the number of tables to 5 or 6 tables

Response 18: We are very grateful to the reviewer for the useful suggestion. We complied with the suggestion and we deleted some tables from the manuscript and added them as supplementary information. Early in this Cover letter we provided an explanation for the tables, because this point was also included in the summary provided by the reviewer in the e-mail sent to the corresponding author. We let in the manuscript only the important tables: the tables with correlations and t tests, and the table with sociodemographic characteristics of the respondents. Thus, we presented again the explanation for the way we included the tables in supplementary information:

We created Word documents with supplementary information for each of our research questions. In this regard in S3_Tables with results to the 1st research question we included Table 1; in S4_Tables with results to the 2nd research question we included Table 4 and Table 5; in S5_Tables with results to the 3rd research question we included Table 6 and Table 8; in S6_Tables with results to the 4th research question we included Table 10 and Table 11; in S7_Tables with results to the 5th research question we included Table 14 and Table 15.

Reviewer 1 point 19: the conclusion section should be summarized to one paragraph summarize your important results and its significance and the future related researches

Response 19: We thank the reviewer very much for the recommendation. We answered to this point previously in this Cover letter, because the same point was also mentioned in the summary provided by the reviewer in the e-mail sent to the corresponding author by the journal (“the conclusion section need to be summarized and conclude the main study findings and its significance.”). In this regard, we present again the redundant information we deleted from the Conclusions section, the way we highlighted the main results, their significance as well as the future research directions.

The information we deleted from the Conclusions section:

“In this regard, besides fighting the pandemic, physician also had to fight the so called infodemic. Fake news spread on social media about various alternative treatments for the virus and the opinions of certain professionals about treatment methods which later proven to be inaccurate negatively influenced the credibility of doctors.” (Lines 789-792)

“This results can suggest that while professionals were aware of the role of social media in spreading medical misinformation and in affecting trust in doctors, due to their knowledge, at personal level they were less affected by that type of information, many of them believing that social media should also be used for sending official information” (lines 803-807)

“Moreover, the medical staff was aware of the alternative treatments which were promoted on social media, the method of drinking alcohol in order to prevent the infection being the method that most of the respondents have heard about” (lines 811-813).

“Hence, on the basis of the findings and implications of the study, we further discuss limitations and future research directions.” (Lines 838-839).

Next, we took into account the recommendation of the reviewer and we started the section by presenting the main findings of our research. Since we had several research questions, we presented our main findings in relation to those research questions. Next, the reviewer recommended us to explain the significance of our study. Thus, in the paper we had already written the theoretical and practical implication of our paper. In this regard, we did not delete the implications because we consider that the implications emphasize why the study conducted is important and how it can be further taken into consideration. Next, we did not delete the limitations and future research directions either, because we considered necessary to highlight how and why our study has limitations but also how it could be further developed or extended. 

Reviewer 1 point 20: please to review your refrences and filter it to 30 to 40 refrences as 83 refrences are too much refrences

Response 20: We are very grateful to the reviewer for the suggestion and we understand the perspective of the reviewer. We would like to mention that we gave an explanation to this point early in this Cover letter, because the point was included in the summary which was sent by e-mail to the corresponding author. However, we insert again below the explanation for this point, explanations in which we show why we were unable to fully comply with the suggestion of the reviewer and delete more than half of our references:

We are very grateful to the reviewer for this recommendation and we appreciated the interest in improving our paper. However, when we started to write the article, we wanted to make sure our paper will be well documented and that it will address all the theoretical concepts and aspects needed. In this regard, we made a thorough research and literature review on the medication used in order to treat the virus, on the way social media contributed to the spread of misinformation about the virus, on the way misinformation influenced people’s confidence in the opinion of doctors and on the way the doctor- patient relation was affected during the pandemic. Thus, we searched and found many research papers and we reviewed all of them because we wanted for our paper to provide an overall view on the subject addressed. In this regard, all the references we used are relevant for the subject approached and for the research that we conducted. In other words, through the references cited we support and sustain our arguments, we show how other researchers approached similar matters and thus we could not afford to reduce them. By reducing them we could no longer have a strong and well consolidated theoretical background and we could not properly explain how we wanted to address the matter of medical misinformation and its effects from the perspective of medical staff. Even more, the journal does not have a limitation regarding the length of the article or the number of references: “Manuscripts can be any length. There are no restrictions on word count, number of figures, or amount of supporting information”. In addition, we have seen articles which addressed subjects related to health and the COVID – 19 pandemic, and which were published in PLOS ONE, that have more than 40 references. For example, one article entitled “Severity of infection with the SARS- CoV -2 B1.1.7 lineage among hospitalized COVID – 19 patients in Belgium” (https://journals.plos.org/plosone/article?id=10.1371/journal.pone.0269138), has 76 references, and another article, entitled “The coronavirus disease 2019 (COVID -19) vaccination psychological antecedent assessment using the ARABIC 5c validated tool: An online survey in 13 Arab countries” (https://journals.plos.org/plosone/article?id=10.1371/journal.pone.0260321) has 71 references.

We would like to mention again that we did our best in trying to address all the suggestions of the reviewer and that we are thankful to the reviewer for all the points raised, for the time spent on analyzing our paper and for providing us very useful recommendations!

Response to reviewer 2

Reviewer 2 comment: The study is interesting and shows the point of view of health professionals, misinformation affected trustwith the patient. Another fact is that even among professionals, there are different perceptions about the spread of fakenews, according to age and occupation. 

Response from authors: We are very grateful to the reviewer for his/hers kind words, and we appreciate the time the reviewer spent on reviewing our paper. We addressed all the recommendations of the reviewer and we will present each of the changes we made to the text. Before describing the way we addressed all the comments, we would like to mention that the changes can be best seen in the revised version of our manuscript, which has the “Track changes” and “All markup” options active.

Reviewer 2 comment 1: Thus, I suggest adequacy in the title, as it is not expressing exactly what the study observed.

Response 1: We thank the reviewer for the very useful suggestion. In order to comply with it, we changed the title of our manuscript in order for it to be more appropriate and more in line with the aim and the results of our study. In this regard, the new title of the manuscript is “Misinformation about medication during the COVID – 19 pandemic – a perspective of medical staff” (Lines 2-3). The title now highlights the fact that the study focused on misinformation about medication during the pandemic, and on the effects that misinformation had on doctors, from the perspective of specialists (doctors, nurses, medical students).

Reviewer 2 comment 2: The survey instrument was validated by a sufficient number of professionals; however, I did not find the attached instrument to be evaluated and to verify that the questions supported the statistical data that was generated. It is important to send supplementary material S1 so that the reviewer can evaluate the work impartially.

Response 2: We are very grateful to the reviewer for pointing this out. However, when we submitted the manuscript, we did upload the questionnaire as supplementary information, both in Romanian language and in English (S1_Appendix English version of the questionnaire; S2_Appendix Romanian version of the questionnaire).| In order to comply with the recommendation of the reviewer, we will try to upload again the questionnaire, and we will also insert it at the end of this document, so that the reviewer can have access to it. In this regard, the reviewer can find below the English and Romanian version of our questionnaire.

Reviewer 2 comment 3: The Information on drugs used to treat COVID 19 topic of the Literature review covers the year 2020 and serves to locatethe context that health professionals were in at the time of answering the questionnaire, however, there is a lack ofinformation on the drugs that were being recommended by the WHO in the period of application of the questionnaire,which was from April to June 2021. Contextualizing how the data were in the period when the instrument was applied can directly impact the conclusion:“Healthcare professionals knew about the drugs used in clinical trials”.

Response 3: We thank the reviewer for the useful suggestion. We searched for sources which contained information regarding the types of drugs available and approved in the period in which we conducted our research (April – June 2021) and we saw that among the drugs approved were also the drugs about which the respondents to our research had knowledge. Besides drugs, the news regarding the virus started to focus also on information about possible vaccines, so the information about antiviral drugs started to be published more rarely. Hence, our conclusion regarding the fact that “Healthcare professionals knew about the drugs used in clinical trials” is still true. Thus, we researched the literature and added an explanation in our Discussion section, but we did not insert the references into our paper, because Reviewer 1 mentioned that we have many references in our paper and that we should reduce them. However, Reviewer 2 can consult the references because we will insert them here after we provide the explanation. Hence, in the Discussion section of our manuscript, page 43, lines 753-760 we added the following explanation:

“Moreover, during the period in which we conducted our research, (April – June 2021), among the drugs which were approved were Remdesivir Tocilizumab – which was authorized first in June 2021, drug which were also acknowledged by the respondents of our research” [Reference 84, Reference 85)].”

Even more, one of the authors of the article (L.R.) is a doctor and was directly involved in the process of taking care of COVID – 19 patients, so the author can confirm that among the drugs which were in trial, or which were approved for administration against COVID-19 were also the drugs which were acknowledged by the respondents of our research. 

Reference 84: Food and drug administration. Coronavirus (COVID-19) Drugs [Internet]. Food and Drug Administration. [cited 2022 June 20] Available from: https://www.fda.gov/drugs/emergency-preparedness-drugs/coronavirus-covid-19-drugs

Reference 85: Murdock, J. The Latest Updates on COVID-19 Treatments and Medications in the Pipeline. [Internet]. 23 May 2022 [cited 2022 June 20] Available from: https://www.goodrx.com/conditions/covid-19/coronavirus-treatments-on-the-way

Reviewer 2 comment 4: Minor revisions: When reading, there are differences in font size/type. E.g. lines 206 and. 534

Response 4: We thank the reviewer for pointing this out. We would firstly like to mention that line 206 has the number 245 in the revised version of the manuscript with “Track changes” and “All markup” option active, and line 534 has the number 645. In order to make sure there will no differences in font/size type, we checked again our manuscript and we corrected the mistakes. In this regard, we made sure the text from our manuscript is all formatted with Calibri, size 12.

We thank again the reviewer for spending time on reviewing our paper and for providing us very useful suggestions!

We are very grateful to the reviewers and the academic editor for all the suggestions, comments and points raised in order to improve our paper!

Sincerely, 

Prof. Dr. Claudiu Coman

---

## [Decision Letter · Decision Letter 2]

30 Aug 2022

PONE-D-22-09134R2Challenges in the communication process during the COVID-19 pandemic- a perspective of medical staffPLOS ONE

Dear Dr. Coman,

Thank you for submitting your manuscript to PLOS ONE. After careful consideration, we feel that it has merit but does not fully meet PLOS ONE’s publication criteria as it currently stands. Therefore, we invite you to submit a revised version of the manuscript that addresses the points raised during the review process.

 Please submit your revised manuscript by Oct 14 2022 11:59PM. If you will need more time than this to complete your revisions, please reply to this message or contact the journal office at plosone@plos.org. Please include the following items when submitting your revised manuscript:A rebuttal letter that responds to each point raised by the academic editor and reviewer(s). You should upload this letter as a separate file labeled 'Response to Reviewers'.A marked-up copy of your manuscript that highlights changes made to the original version. You should upload this as a separate file labeled 'Revised Manuscript with Track Changes'.An unmarked version of your revised paper without tracked changes. You should upload this as a separate file labeled 'Manuscript'.If applicable, we recommend that you deposit your laboratory protocols in protocols.io to enhance the reproducibility of your results. Protocols.io assigns your protocol its own identifier (DOI) so that it can be cited independently in the future. For instructions see: https://journals.plos.org/plosone/s/submission-guidelines#loc-laboratory-protocols. Additionally, PLOS ONE offers an option for publishing peer-reviewed Lab Protocol articles, which describe protocols hosted on protocols.io. Read more information on sharing protocols at https://plos.org/protocols?utm_medium=editorial-email&utm_source=authorletters&utm_campaign=protocols.

We look forward to receiving your revised manuscript.

Kind regards,

Markus Ries, MD PhD MHSc FCP

Academic Editor

PLOS ONE

Journal Requirements:

Reviewers' comments:

Reviewer's Responses to Questions

**Comments to the Author**

1. If the authors have adequately addressed your comments raised in a previous round of review and you feel that this manuscript is now acceptable for publication, you may indicate that here to bypass the “Comments to the Author” section, enter your conflict of interest statement in the “Confidential to Editor” section, and submit your "Accept" recommendation.

Reviewer #1: All comments have been addressed

Reviewer #2: All comments have been addressed

2. Is the manuscript technically sound, and do the data support the conclusions?

Reviewer #1: Partly

Reviewer #2: Yes

3. Has the statistical analysis been performed appropriately and rigorously? 

Reviewer #1: Yes

Reviewer #2: Yes

4. Have the authors made all data underlying the findings in their manuscript fully available?

Reviewer #1: Yes

Reviewer #2: Yes

5. Is the manuscript presented in an intelligible fashion and written in standard English?

Reviewer #1: No

Reviewer #2: Yes

6. Review Comments to the Author

Reviewer #1: Thanks for fixing the required modifications and please find the required minor modifications for improving the quality of the research paper.

Reviewer #2: The article has improved considerably and it is more accurate in the information.

The article brings relevant data on the medical perspective in the face of Fake news.

Lines 404, 474, 525 need to be revised because it has some formatting problem.

7. PLOS authors have the option to publish the peer review history of their article (what does this mean?). If published, this will include your full peer review and any attached files.

Reviewer #1: **Yes: **Professor. Dr. Sally Mohammed Farghaly

Reviewer #2: **Yes: **LARISSA DEADAME DE FIGUEIREDO NICOLETE

---

## [Author Response · Author response to Decision Letter 2]

22 Sep 2022

Our response to Reviewer 1: 

We firstly thank the reviewer for taking time to review our manuscript again and provide suggestions in order to improve it. We addressed all the suggestions made by the reviewer. When we described how the text was changed, we also provide the lines where the text can be found in the revised manuscript with the option “Track changes” active. In this way, the changes can be seen completely (the text we deleted, and the text we inserted). Next, we present our response to the comments made by the reviewer in the PDF version of our manuscript. 

Reviewer 1 comments- made in the PDF version of our revised manuscript 

Reviewer 1 comment 1: Research design need to be separated in a separate line.

Response 1: We thank the reviewer for the suggestion. At line 307 in our revised manuscript the expression “Research design” was next to the title Materials and methods. In order to comply with the suggestion of the reviewer we put the expression “Research design” in a separate line below the title. In this regard, the change can be seen while having active the “Track changes” function and “All markup” option from Microsoft Word, at lines 307-308.

Reviewer 1 comment 2: what type of qualitative research design ?

Response 2: We thank the reviewer for pointing this out. However, we specified in the research design section that we did not use qualitative research design, but quantitative research design. The type of quantitative research design we used is descriptive. In this regard, we added some information in the subsection Research design, and we explained that we used descriptive quantitative research design. The change can be seen at line 310 in the revised manuscript with “Track changes” and “All Markup” function active:

“The method used is quantitative and descriptive”

Reviewer 1 comment 3: please to review and confirm that type of sampling method as it is used usually wit the qualitative data collection.

Response 3: We are thankful to the reviewer for pointing that we should check again if the sampling method is appropriate for the type of research design we used. We checked and reviewed again the type of sampling and we, the authors, made sure that the type of sampling method is appropriate for the research design we used, which is quantitative, not qualitative.

Reviewer 1 comment 4: the tables should be presented in sequence with numbering which is different from the supplementary tables numbering. please to review and fix it.

Response 4: We are grateful to the reviewer for the very useful suggestion. We made sure that the tables are numbered in sequence. In this regard, we renamed all the tables from our supplementary files in order for them not be confused with the numbers of the tables within the manuscript. We further describe each change that we made to the text while having active the “Track changes” function and the “All markup” option:

Because some data was missing from table 2 in our revised manuscript, we inserted again Table 2 within the text. The table was initially put into supplementary files, S3 Tables with results to the 1st research question, but since we inserted it into the text again, we deleted the supplementary file called S3 Tables with results to the 1st research question. The insertion of the table can be found at line 406, page 20 of the revised manuscript and the change of deleting the supplementary file can be seen at line 1007 of the revised manuscript with “Track changes” function and “All Markup” option active.

Next, 

Lines 452-453: S4_ Tables with results to the 2nd research question_Table 5 – became S3_ Tables with results to the 2nd research question_Table A

Lines 462 – 462: S4_ tables with results to the 2nd reseach question_Table 6 – became ( S3_ tables with results to the 2nd reseach question_Table B

Next, from the supplementary file entitled (S4_ Tables with results to the 3rd research question – we deleted the table that had the number 7 in our revised manuscript and we inserted it within the text. The inserted table can be found at line 477 in our revised manuscript, page 25.

Next, the table that had the number 8, now has the number 6 in order for the tables within our manuscript to be numbered in sequence. The change can be seen after the line 484.

Then, at lines 490- 491: S5_ Tables with results to the 3rd research question_Table 9 became S4_ Tables with results to the 3rd research question_Table C.

Then, the table that had the number 10, now has the number 7 in order for the tables within our manuscript to be numbered in sequence. The change can be seen at line 504.

Then, from S5 Tables with results to the 4th research question_ we deleted the table that had the number 11 in the supplementary file and we inserted it again into the manuscript. In this regard, the table received the number 8, in order for our tables to be numbered in sequence. The inserted table can be found at line 529 in our manuscript with “Track changes” and “All markup” option active.

Then, at line 534, S6 Tables with results to the 4th research question_Table 12 became S5 Tables with results to the 4th research question_Table D. 

Then, the table that had the number 13, now has the number 9 in order for the tables within our manuscript to be numbered in sequence. The change can be seen at line 548.

Then, the table that had the number 14, now has the number 10 in order for the tables within our manuscript to be numbered in sequence. The change can be seen at line 562.

Then, at lines 573-574: S7 Tables with results to the 5th research question_Table 15 became S6 Tables with results to the 5th research question_Table E.

Then, at lines 580 – 581: S7 Tables with results to the 5th research question_Table 16 became S6 Tables with results to the 5th research question_Table F.

In this regard, due to the fact that we inserted into the manuscript some of the tables that were in the supplementary files in order for our tables to be numbered in sequence, our supplementary files changed:

S3 Tables with results to the 1st research question – was deleted completely

S4 Tables with results to the 2nd research question_ became S3 Tables with results to the 2nd research question

S5 Tables with results to the 3rd research question_ became S4 Tables with results to the 3rd research question_

S6 Tables with results to the 4th research question_ became S5 Tables with results to the 4th research question_

S7 Tables with results to the 5th research question_ became S6 Tables with results to the 5th research question 

The changes can be seen at lines 1007 – 1011 in our revised manuscript with “Track changes” and “All markup” options active.

Reviewer 1 comment 5: is there any missed data?

Response 5: We are very grateful to the reviewer for pointing out that some data was missing from our tables. The reviewer was referring to Table 2 which Was at line 404 in the PDF version of our manuscript. We inserted again the table with all the data available. The inserted table can be seen between lines 406 – 407 with “Track changes” function and “All markup” option active.

Table 2. The extent to which information about alternative treatments affected trust in physicians

 Frequency Percent Valid Percent Cumulative Percent

Valid to an extremely little extent 14 2.6 2.6 2.6

 to a very little extent 10 1.9 1.9 4.5

 to a little extent 42 7.8 7.8 12.3

 nor to a little, neither to a great extent 58 10.8 10.8 23.1

 to a great extent 114 21.3 21.3 44.4

 to a very great extent 124 23.1 23.1 67.5

 to an extremely great extent 174 32.5 32.5 100.0

 Total 536 100.0 100.0 

Reviewer 1 comment 6: is there any missed data?

Response 6: We are very grateful to the reviewer for pointing out that some data was missing from our tables. The reviewer was referring to the table which had the number 7 in our revised manuscript, which Was at line 474 in the PDF version of our manuscript. We inserted again the table with all the data available. The inserted table now has the number 5 so that our tables can be numbered in sequence and the table can be seen between lines 477 – 478 with “Track changes” function and “All markup” option active

Table 5. The level of satisfaction with the way information about drugs used to treat the virus were communicated at national level 

 Frequency Percent Valid Percent Cumulative Percent

Valid extremely dissatisfied 52 9.7 9.7 9.7

 very dissatisfied 76 14.2 14.2 23.9

 dissatisfied 110 20.5 20.5 44.4

 Nor dissatisfied, neither satisfied 136 25.4 25.4 69.8

 satisfied 108 20.1 20.1 89.9

 very satisfied 30 5.6 5.6 95.5

 Extremely satisfied 24 4.5 4.5 100.0

 Total 536 100.0 100.0 

Reviewer 1 comment 7: is there any missed dara?

Response 7: We are very grateful to the reviewer for pointing out that some data was missing from our tables. The reviewer was referring to the table which Had the number 11 in our revised manuscript, which was at line 525 in the PDF version of our manuscript. We inserted again the table with all the data available. The inserted table now has the number 8 so that our tables can be numbered in sequence and the table can be seen between lines 429 – 430 with “Track changes” function and “All markup” option active

Table 8. Perception about the extent to which social media contributed to the spread of medical fake news

 Frequency Percent Valid Percent Cumulative Percent

Valid to an extremely little extent 2 .4 .4 .4

 to a very little extent 10 1.9 1.9 2.2

 to a little extent 12 2.2 2.2 4.5

 nor to a little, neither to a great extent 30 5.6 5.6 10.1

 to a great extent 62 11.6 11.6 21.6

 to a very great extent 88 16.4 16.4 38.1

 to an extremely great extent 332 61.9 61.9 100.0

 Total 536 100.0 100.0 

Reviewer 1 comment 8: where is table 15?

Response 8: We thank the reviewer for the question. Table 15 was inserted as supplementary information. Due to the fact that we changes the numbers of the tables in order for the table within our manuscript to not be confused with the ones from the supplementary files, Table 15 is now Table E and can be found in the file entitled: S6 Tables with results to the 5th research question_Table E.

We are thankful to the reviewer for all the points raised, for the time spent on reviewing our paper and for providing us very useful suggestions!

Our response to Reviewer 2: 

We are grateful to the reviewer for all the useful suggestions and we appreciate the time the reviewer spent on reviewing our paper. We addressed all the recommendations of the reviewer and we will present each of the changes we made to the text. Thus, we would like to mention that the changes can be best viewed in the revised version of our manuscript, which has the “Track changes” and “All markup” options active.

Reviewer 2 comment 1 – as found in the decision letter received by e-mail by the corresponding author: Lines 404, 474, 525 need to be revised because it has some formatting problem.

Response 1: We thank the reviewer for the very useful suggestion. In order to comply with it we checked the format of all the lines mentioned by the reviewer and we tried to fix them. 

We thank again the reviewer for the kind words and for all the time she spent on analyzing our paper!

We are very grateful to the reviewers and to the academic editor for all the suggestions, comments and points raised in order to improve our paper!

Sincerely, 

Prof. Dr. Claudiu Coman

---

## [Decision Letter · Decision Letter 3]

12 Oct 2022

Misinformation about medication during the COVID -19 pandemic: a perspective of medical staff

PONE-D-22-09134R3

Dear Dr. Coman,

We’re pleased to inform you that your manuscript has been judged scientifically suitable for publication and will be formally accepted for publication once it meets all outstanding technical requirements.

Kind regards,

Markus Ries, MD PhD MHSc FCP

Academic Editor

PLOS ONE

Additional Editor Comments (optional):

This revision addresses the issues raised by the two reviewers in the last round. There are still some typos in the manuscript. Please correct these in the next steps for publication.

Reviewers' comments:

Reviewer's Responses to Questions

**Comments to the Author**

1. If the authors have adequately addressed your comments raised in a previous round of review and you feel that this manuscript is now acceptable for publication, you may indicate that here to bypass the “Comments to the Author” section, enter your conflict of interest statement in the “Confidential to Editor” section, and submit your "Accept" recommendation.

Reviewer #1: All comments have been addressed

2. Is the manuscript technically sound, and do the data support the conclusions?

Reviewer #1: Yes

3. Has the statistical analysis been performed appropriately and rigorously? 

Reviewer #1: Yes

4. Have the authors made all data underlying the findings in their manuscript fully available?

Reviewer #1: Yes

5. Is the manuscript presented in an intelligible fashion and written in standard English?

Reviewer #1: Yes

6. Review Comments to the Author

Reviewer #1: (No Response)

7. PLOS authors have the option to publish the peer review history of their article (what does this mean?). If published, this will include your full peer review and any attached files.

Reviewer #1: No

---

## [Editor Report · Acceptance letter]

18 Oct 2022

PONE-D-22-09134R3 

Misinformation about medication during the COVID – 19 pandemic: a perspective of medical staff 

Dear Dr. Coman:

I'm pleased to inform you that your manuscript has been deemed suitable for publication in PLOS ONE. Congratulations! Your manuscript is now with our production department. 

Kind regards, 

on behalf of

Professor Markus Ries 

Academic Editor

PLOS ONE